# Your Models Have Thought Enough: Training Large Reasoning Models to Stop Overthinking

**Jinyi Han**[♡]**, Ying Huang** [◇]**, Ying Liao**[◇]**, Zishang Jiang** [◇]**, Xikun Lu** [♡]**, Haiquan Zhao**[♠]**,
Xinyi Wang**[◇]**, Guanghao Zhou**[△]**, Sihang Jiang** [♠]**, Weikang Zhou** [♣]**,
Zeye Sun** [♣]**, Fei Yu** [♣]**, Yanghua Xiao**[♡, ♠*]**, Jiaqing Liang** [◇] [*]

[♡]Shanghai Institute of Artificial Intelligence for Education, East China Normal University
[◇]School of Data Science, Fudan University
[△]School of Data Science and Engineering, East China Normal University
[♠]College of Computer Science and Artificial Intelligence, Fudan University [♣] Ant Group

## Abstract

Large Reasoning Models (LRMs) have achieved impressive performance on challenging tasks, yet their deep reasoning often incurs substantial computational costs. To achieve efficient reasoning, existing reinforcement learning methods still struggle to construct short reasoning path during the rollout stage, limiting effective learning. Inspired by Evidence Accumulation Models, we find that LRMs have accumulated sufficient information early in reasoning, making further reasoning steps redundant. Based on this insight, we propose Just-Enough Thinking (JET), which trains models to proactively terminate unnecessary reasoning. JET performs trajectory truncation during rollout to expose the model to short, distributionally consistent reasoning paths. Besides, it uses a quality-controlled length reward to better encourage concise reasoning while maintaining correctness. Extensive experiments demonstrate that JET significantly improves reasoning efficiency without sacrificing accuracy. In particular, JET delivers a 4.6% accuracy improvement while reducing the output length by 46.3% on the Olympiad benchmark using DeepSeek-R1-Distill-Qwen-1.5B. Our code is available in the GitHub [1].

## 1 Introduction

Large Reasoning Models (LRMs) have achieved substantial performance gains on highly structured and complex reasoning tasks, such as mathematical problem solving (Shao et al., 2024) and competitive programming (Jiang et al., 2024). Their reasoning process involves elaborate intermediate steps, reflective self-verification, and exploring alternative solution strategies (Min et al., 2024). These developments are widely regarded as an approximation of human System-2 cognitive processes (Qu et al., 2025a; Li et al., 2025). Models including OpenAI o1 (OpenAI, 2025) and DeepSeek-R1 (Guo et al., 2025) empirically demonstrate this capability in practice.

Despite these advances, the System-2 style reasoning of LRMs incurs substantial computational costs. LRMs frequently perform more reasoning steps than necessary and consuming excessive computational resources to reach correct answers, a phenomenon we refer to as "overthinking" (Chen et al., 2024; Luo et al., 2025b). This contrasts sharply with the efficiency and adaptability of human cognition. Therefore, developing methods that maintain high accuracy while improving computational efficiency has become a central challenge for LRMs.

To address the challenge of reasoning efficiency, Reinforcement Learning (RL) has emerged as a promising paradigm (Hu et al., 2025b; Liu et al., 2025b). The core idea is to *use additional reward signals to guide model generation toward answers that are both correct and concise*. Existing approaches fall into two main categories. (i) adaptive thinking mode selection methods equip models with multiple reasoning modes (think/no-think) via Supervised Fine-Tuning (SFT), and then use RL with thinking rewards to select the most suitable mode for each problem (Zhang et al., 2025; Wu

---

[*]Corresponding authors
[1]https://github.com/JinyiHan99/Just-Enough-Think/

et al., 2025a; Huang et al., 2025a). (ii) length-based optimization approaches introduce explicit length rewards to encourage more concise reasoning (Team et al., 2025; Gao et al., 2025).

Effective reward-based methods depend on exposing the model to diverse samples, yet LRMs naturally favor verbose outputs, rarely generating short reasoning trajectories. This inherent verbosity biases the training data, as shown in the Figure 1a, leading to a flawed reward signal that fails to encourage concise reasoning (Wu et al., 2025b). A straightforward solution is to compress long answers or provide external short answers. Yet, such interventions introduce a **significant distribution mismatch** between the model's natural generation distribution and the artificially shortened samples, which destabilizes gradient updates and impairs learning within the model's own generative process (Huang et al., 2025b).

To overcome these limitations, it's necessary to identify **short, distributionally consistent** reasoning trajectories from the model's own long reasoning chains. Inspired by Evidence Accumulation Models (Lee & Cummins, 2004) in cognitive science, which describe that human decision-making is a dynamic process where information is integrated until a threshold is reached, after which further evidence serves only to support the decision. We hypothesize that LRM reasoning operates similarly. The early part of a reasoning trajectory already contains enough information to determine the final answer, and continues to generate results in redundant reasoning. Our pilot experiments also support this hypothesis.

Building on this core insight, we propose Just-Enough Thinking (JET), a method that trains models to proactively terminate unnecessary thinking. Specifically, JET operates through two key components. First, we additionally perform trajectory truncation during RL rollout, exposing the model to reasoning paths of varying lengths while keeping them aligned with the model's natural generation. Second, we introduce a quality-controlled length reward to better guide the model toward efficiency. For each reasoning group, if multiple trajectories reach the correct answer, we use the shortest correct trajectory as a reference and assign a length reward to all correct trajectories, where the reward decreases gradually with trajectory length.

Our main contributions are as follows:

• Inspired by the Evidence Accumulation Models, we propose Just-Enough Thinking (JET), which leverages trajectory truncation to expose the model to reasoning paths of different lengths during rollout, and length-aware rewards to guide more efficient reasoning.

• We show empirically that LRMs accumulate most of the necessary information early in the reasoning process. Truncating rollouts allows us to construct short reasoning paths that stay aligned with the model's natural generation distribution, providing a principled basis for training JET.

• We provide a theoretical analysis showing that our two-stage truncation strategy preserves effective learning signals, allowing the policy model to learn reliably from shortened reasoning trajectories.

• Extensive experiments demonstrate that JET delivers strong gains in efficient reasoning on complex tasks. On Olympiad, for example, JET improves accuracy by 4.6% while reducing output length by 46.3% on DeepSeek-Distill-Qwen-1.5B. The model's proactive stop-thinking behavior also generalizes well to other reasoning benchmarks.

## 2 PILOT EXPERIMENTS

LRMs often produce long multi-step reasoning sequences, but it remains unclear whether the full sequence is necessary to arrive at the correct answer. In this section, we investigate **whether the model accumulates sufficient information during the initial reasoning steps to produce correct answers.** Therefore, we conduct pilot experiments on the MATH500 dataset with DeepSeek-Distill-Qwen-7B. We mainly evaluate how limiting the model to only the early portion of its reasoning sequence affects answer correctness.

### 2.1 TASK DEFINITION

Let $M$ be a large reasoning model. For a given problem $q \in \mathcal{Q}$, $M$ generates a sequence of intermediate reasoning states $S = [s_1, s_2, \ldots, s_L]$ and produces a final answer $o_q$, where $L$ denotes

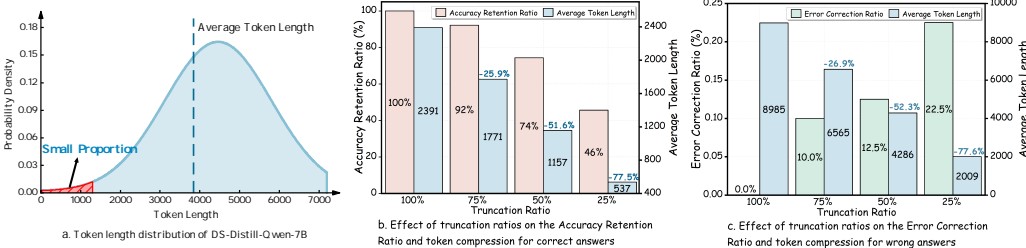

Figure 1: **a.** The token length distribution of 500 answers generated by DeepSeek-R1-Distill-Qwen-7B on a math problem. Answers shorter than 1,000 tokens are extremely rare, showing that LRMS hard to produce short answers on their own. **b.** The effect of truncation ratios on the Accuracy Retention Ratio and token compression for the DeepSeek-R1-Distil-Qwen-7B model on the MATH500 dataset. **c.** The effect of truncation ratios on the Error Correction Ratio and token compression for wrong answers.

the total length of reasoning steps. Following the standard autoregressive generation paradigm, this process is formulated as:

$$P\big(S_q, o_q \mid q\big) = \prod_{i=1}^{L} P\big(s_i \mid q, s_{<i}\big) \cdot P\big(o_q \mid q, S_q\big). \tag{1}$$

We define a **truncated reasoning sequence** $S_q^{(T)}$ as the first $\lfloor T \cdot L \rfloor$ intermediate reasoning process of the full sequence, where $S_q^{(T)} = [s_1, s_2, \ldots, s_{\lfloor T \cdot L \rfloor}], T \in [0, 1)$. At the truncation point, a forced **stop-thinking signal** $z_{\text{stop}}$ is inserted, instructing the model to stop thinking and produce a final answer $\hat{A}^{(T)}$ based on the available intermediate reasoning steps.

$$\hat{o}_q^{(T)} \sim P(o_q \mid q, S_q^{(T)}, z_{\text{stop}}), \tag{2}$$

To quantify the model's performance under truncation, we use three complementary metrics that capture correctness and efficiency:

*Accuracy Retention Ratio (ARR).* It measures the proportion of originally correct problems that remain correct after truncation. Let $\mathcal{Q}_{\text{correct}} = \{ q \in \mathcal{Q} \mid o_q = o_q^* \}$ be the set of problems correctly solved with the full reasoning sequence, where $o_q^*$ is the ground-truth answer. Then

$$\text{ARR}(T) = \frac{1}{|\mathcal{Q}_{\text{correct}}|} \sum_{q \in \mathcal{Q}_{\text{correct}}} \mathbb{I}\big(\hat{o}_q^{(T)} = o_q^*\big). \tag{3}$$

*Error Correction Rate (ECR).* It measures the proportion of initially incorrect problems that become correct after truncation. Let $\mathcal{Q}_{\text{incorrect}} = \{ q \in \mathcal{Q} \mid o_q \neq o_q^* \}$ be the set of problems that are incorrectly solved with the full reasoning sequence. Then

$$\text{ECR}(T) = \frac{1}{|\mathcal{Q}_{\text{incorrect}}|} \sum_{\bar{q} \in \mathcal{Q}_{\text{incorrect}}} \mathbb{I}\big(\hat{o}_{\bar{q}}^{(T)} = o_q^*\big), \tag{4}$$

where $\hat{o}_{\bar{q}}^{(T)}$ is the answer after truncating the reasoning sequence.

*Length Compression Ratio.* It quantifies the reduction in reasoning tokens achieved by relying solely on the truncated prefix to produce the final answer.

$$\text{LCR}(T) = 1 - \frac{|S_q^{(T)}|}{|S_q|} \tag{5}$$

## 2.2 PILOT EXPERIMENTS AND ANALYSIS

We first establish a baseline by allowing the model to generate complete reasoning traces and its final answers. We then truncate the reasoning process, retaining only the first 75%, 50%, and 25% of the original reasoning chain, and insert a stop-thinking cue, like *"Wait, I have enough information to*

*get the final answer. Therefore, the final answer is..."* (as shown in the left of Figure 2), to prompt the model to produce the final answer from the partial reasoning. We evaluate its performance under these conditions.

LRMs retain most of their accuracy even when large portions of the reasoning chain are removed. As shown Figure 1b, restricting the model to the first 75% of its reasoning preserves over 90% of the originally correct solutions, and using only the first half still yields correct answers on roughly three-quarters of those problems. Even with just a quarter of the reasoning, nearly half of the originally correct solutions survive. This indicates that the essential problem-solving information is accumulated early and later steps contribute little to correctness.

Furthermore, truncating reasoning trajectories can turn some initially incorrect answers into correct ones, effectively mitigating the "overthinking" problem. Figure 1c shows that progressively shortening the reasoning chain increases the fraction of previously incorrect solutions that are corrected. Redundant or excessive reasoning steps sometimes mislead the LRMs. Limiting the reasoning process appropriately improves overall answer accuracy.

Truncation also yields substantial reductions in token consumption. Full reasoning requires an average of about 2,391 tokens per problem. Retaining only the first three-quarters of the reasoning chain reduces token consumption by roughly 25% with almost no accuracy loss. Even when using only the first 25% of the chain, token usage drops to about 75% of the original. Although accuracy decreases at this point, the model still delivers reasonable performance at a significantly lower computation cost.

> **Highlights**
>
> LRMs accumulate most of the crucial problem-solving information early in their reasoning process, allowing them to maintain high accuracy while significantly reducing the number of reasoning tokens.

## 3 METHODS

Motivated by our observation that LRMs accumulate most problem-solving information in the early stages of reasoning, we introduce JET, a reinforcement learning approach based on DAPO (Yu et al., 2025). JET trains LRMs to stop reasoning once sufficient information has been gathered, enabling them to produce accurate answers and concise reasoning steps.

### 3.1 TWO-STAGE ROLLOUT CONSTRUCTION

To expose the model to diverse reasoning behaviors and facilitate efficient policy learning, we propose a two-stage rollout construction strategy, illustrated in Figure 2. This approach generates both full and truncated reasoning trajectories, enabling the model to balance correctness and brevity during reinforcement learning.

**Stage1: Full Reasoning**   In the first stage, the model generates complete reasoning trajectories through its standard autoregressive process. These full trajectories capture all intermediate steps and the final answer, providing a comprehensive reference of the model's natural reasoning behavior. They serve as a foundation for constructing truncated trajectories in the next stage and allow sampling of diverse reasoning paths for RL rollouts.

**Stage2: Trajectory Truncation**   This stage constructs shorter reasoning paths by truncating full trajectories. Building on the full trajectories from Stage 1, we construct shorter reasoning paths by truncating at various intermediate steps. At each truncation point, we insert an explicit *stop-thinking* cue that instructs the model to immediately output a conclusion, rather than continuing further. This ensures that truncated trajectories remain consistent with the model's generation distribution while introducing diverse early-stopped reasoning paths.

**Identify the Truncation Position.** Determining the optimal truncation point is non-trivial. truncating too early may produce incorrect answers, while truncating too late adds redundant steps. Exhaustive search is computationally costly and reduces RL efficiency. To address this, we introduce Progressive Early-Stopping (PES), which generates a sequence of truncated variants along each full trajectory,

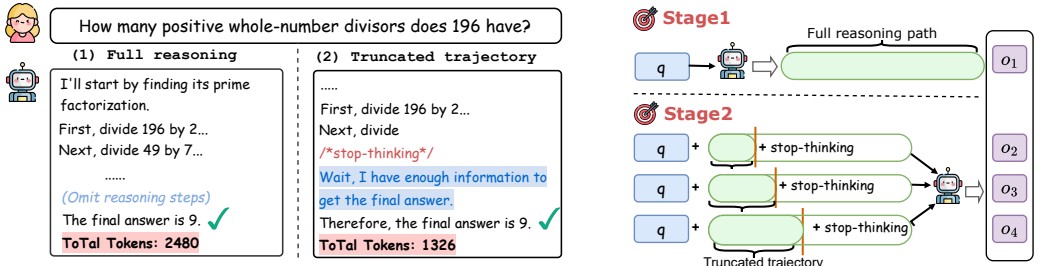

Figure 2: **Left:** An example of a truncated reasoning trajectory; **Right:** The process of Two-stage Rollout Construction.

formalized as:

$$t_k = t_0 + k\Delta t, \quad k = 0, 1, \ldots, K \tag{6}$$

where $t_0$ is the initial cutoff, $\Delta t$ is the predefined step size or quantile interval, and $K$ controls the number of truncations. At each cutoff $t_k$, an explicit stop-thinking cue prompts the model to immediately produce its final answer.

This progressive truncation strategy (i) preserves consistency with the model's own generation distribution, (ii) likelihood of capturing an optimal or near-optimal cutting point $t^*$, and (iii) provides diverse early-stopped trajectories that guide the model to learn when to halt reasoning effectively.

Together, the two stages generate diverse reasoning trajectories that guide the RL policy to reason efficiently without sacrificing accuracy. Full trajectories provide comprehensive coverage, while truncated trajectories encourage concise reasoning, enabling the model to balance correctness and brevity.

## 3.2 REWARD AND OBJECTIVE DESIGN

Another critical component in RL training is the design of reward system, which guides the model toward desired behaviors. Our reward mechanism consists of two main components: a base reward that encourages correct and well-formatted answers, and a length reward that encourages concise reasoning.

**Base Reward.** Following standard RL practices, the base reward combines two components: the format reward $r_f$ and the correctness reward $r_{acc}$. The format reward $r_f \in \{0, 1\}$ ensures proper answer presentation by requiring final answers to be enclosed in \boxed{} tags. The correctness reward $r_{acc} \in \{0, 1\}$ evaluates answer accuracy through exact string matching with the ground-truth solution.

**Length Reward.** To encourage concise reasoning without compromising correctness, we design an *accuracy-conditioned length reward* based on the following principles:

*Correctness first:* Only correct responses are eligible for length rewards, ensuring that accuracy remains the primary objective.
*Conciseness preference:* Among correct responses, shorter reasoning paths receive higher rewards, promoting brevity in reasoning.
*Per-question normalization:* Rewards are measured relative to the shortest and longest correct responses for each question, preventing biases caused by differing length distributions across questions.

Formally, let $\mathcal{C} = \{i \mid r_{acc}(i) = 1\}$ denote the set of correct responses for a question. Within this set, we define $\ell_{\min} = \min_{j \in \mathcal{C}} \ell_j$ and $\ell_{\max} = \max_{j \in \mathcal{C}} \ell_j$ as the lengths of the shortest and longest correct responses, respectively. The length reward for a response $i$ is then:

$$r_\ell(i) = \begin{cases} \left( \frac{\ell_{\max} - \ell_i}{\ell_{\max} - \ell_{\min} + \varepsilon} \right) \cdot \alpha \cdot (1 - \delta) + \delta, & \text{if } i \in \mathcal{C} \\ 0, & \text{if } i \notin \mathcal{C} \end{cases} \tag{7}$$

Here, $\alpha$ controls the rate at which reward decays with length, $\delta \in (0, 1)$ sets a minimum reward for the correct responses, and $\varepsilon > 0$ avoids division by zero when all correct responses have equal length.

**Total Reward.** The total reward combines the contributions of correctness, formatting, and conciseness to guide the model toward accurate, well-formatted, and concise reasoning. Formally, for a sample $i$, the total reward is defined as:

$$R(i) = w_f \cdot r_f(i) + w_{acc} \cdot r_{acc}(i) + w_\ell \cdot r_\ell(i) \tag{8}$$

where $r_f(i)$, $r_{acc}(i)$, and $r_\ell(i)$ are the format, correctness, and length rewards, respectively, and $w_f$, $w_{acc}$, and $w_\ell$ are non-negative coefficients that balance their contributions. By combining these components, the reward function ensures that the model not only produces correct answers but also maintains proper formatting and favors concise reasoning paths.

**Training Objective.** The original DAPO algorithm optimizes the policy by sampling a set of outputs $\{o_i\}_{i=1}^G$ for each input query $q$ and corresponding answer $A$, with the objective formulated as:

$$\mathcal{J}_{\text{DAPO}}(\theta) = \mathbb{E}_{(q,a)\sim\mathcal{D},\{o_i\}_{i=1}^G \sim \pi_{\theta_{\text{old}}}(\cdot|q)}$$

$$\left[ \frac{1}{\sum_{i=1}^G |o_i|} \sum_{i=1}^G \sum_{t=1}^{|o_i|} \min\left( r_{i,t}(\theta)\hat{A}_{i,t}, \ \text{clip}\left( r_{i,t}(\theta), 1-\varepsilon_{\text{low}}, 1+\varepsilon_{\text{high}} \right)\hat{A}_{i,t} \right) \right] \tag{9}$$

Here, the importance sampling ratio is $r_{i,t}(\theta) = \frac{\pi_\theta(o_{i,t}|q,o_{i,<t})}{\pi_{\theta_{\text{old}}}(o_{i,t}|q,o_{i,<t})}$, and the advantage estimate is $\hat{A}_{i,t} = \frac{R_i - \text{mean}(\{R_i\}_{i=1}^G)}{\text{std}(\{R_i\}_{i=1}^G)}$.

Unlike standard DAPO, which computes the loss only on full reasoning trajectories, JET incorporates both full trajectories and those obtained by truncating a reasoning chain and then completing it. By including these truncated-and-completed trajectories in the objective, the policy learns to stop reasoning once sufficient information has been gathered, producing answers that are both correct and concise. The complete algorithm of JET is shown in Appendix 1.

### 3.3 Gradient Analysis of Truncated Rollouts in JET

In JET, the truncation operation appends a short stop-thinking sentence followed by the final answer. This operation does not introduce significant discontinuities in the sequence-level probability distribution and thus preserves the stability of RL updates. Here, we analyze its effect under the DAPO objective (Eq. 9).

Consider a trajectory $\hat{o}_i^{(\mathcal{T})} = \{o_{i,1}, \ldots, o_{i,\mathcal{T}+m}\}$, where the first $\mathcal{T}$ tokens $\{o_{i,1}, \ldots, o_{i,\mathcal{T}}\}$ are sampled from the old policy $\pi_{\theta_{\text{old}}}$, and the last $m$ tokens $\{o_{i,\mathcal{T}+1}, \ldots, o_{i,\mathcal{T}+m}\} = \{\varphi_{\mathcal{T}+1}, \ldots, \varphi_{\mathcal{T}+m}\}$ form a *forced suffix* with $m \ll \mathcal{T}$. The log-probability of the full trajectory under the current policy $\pi_\theta$ factorizes autoregressively as

$$\log \pi_\theta(\hat{o}_i) = \sum_{t=1}^{\mathcal{T}} \log \pi_\theta(o_{i,t} \mid o_{i,1:t-1}) + \sum_{t=\mathcal{T}+1}^{\mathcal{T}+m} \log \pi_\theta(\varphi_t \mid o_{i,1:t-1}). \tag{10}$$

Equivalently, it is represented as

$$\pi_\theta(\hat{o}_i) = \pi_\theta(o_{i,1:\mathcal{T}}) \cdot C, \quad C = \prod_{t=\mathcal{T}+1}^{\mathcal{T}+m} \pi_\theta(\varphi_t \mid o_{i,1:t-1}). \tag{11}$$

Since $m \ll \mathcal{T}$, the suffix factor $C$ contributes negligibly to the overall trajectory probability.

The stability of RL updates depends on token-level importance ratios $r_{i,t}(\theta)$. For prefix tokens ($t \leq \mathcal{T}$) sampled from $\pi_{\theta_{\text{old}}}$, the trust-region constraint ensures $\pi_\theta \approx \pi_{\theta_{\text{old}}}$, so that $r_{i,t}(\theta) \approx 1$ and the ratios remain within the clipping bounds. For the forced suffix tokens ($t > \mathcal{T}$), although the policy may assign different probabilities, their impact is limited because (i) their importance ratios are clipped to prevent extreme updates, and (ii) their total number $m$ is small.

Consequently, assuming the forced suffix has non-negligible probability under the old policy, the sequence-level importance ratio satisfies

$$\frac{\pi_\theta(\hat{o}_i)}{\pi_{\theta_{\text{old}}}(\hat{o}_i)} = \prod_{t=1}^{\mathcal{T}} r_{i,t}(\theta) \cdot \prod_{t=\mathcal{T}+1}^{\mathcal{T}+m} r_{i,t}(\theta) \approx 1. \tag{12}$$

This shows that appending a short forced suffix preserves the sequence-level distribution and ensures stable policy updates within the trust region.

Table 1: Performance of different baselines across various math tasks. Values in parentheses under *ACC* indicate the accuracy change $\Delta acc$ relative to the *Base*. For each benchmark, the value under *Length* reports the mean token compression ratio across all samples in that benchmark. The value under the *AVG* column further averages these per-benchmark compression ratios to summarize the overall compression performance.

| Methods | GSM8K | | MATH500 | | AIME24 | | AMC | | Olympiad | | AVG | |
|---|---|---|---|---|---|---|---|---|---|---|---|---|
| | ACC | Length | ACC | Length | ACC | Length | ACC | Length | ACC | Length | ACC | Length |
| **DeepSeek-R1-Distill-Qwen-1.5B** | | | | | | | | | | | | |
| Base | 76.0 | 468 | 79.6 | 3617 | 28.7 | 11046 | 63.3 | 7644 | 47.0 | 7679 | 58.9 | 4765 |
| SFT | 81.4 (+5.4) | 559 (+19.4) | 78.8 (-0.8) | 2591 (-28.4) | 27.7 (-1.0) | 9139 (-17.3) | 57.2 (-6.1) | 5394 (-29.4) | 42.5 (-4.5) | 5532 (-28.0) | 57.5 (-1.4) | 3583 (-16.7) |
| DPO | 80.2 (+4.2) | 530 (+13.2) | 78.6 (-1.0) | 2652 (-26.7) | 24.0 (-4.7) | 9966 (-9.8) | 59.0 (-4.3) | 5482 (-28.3) | 44.0 (-3.0) | 5929 (-22.8) | 57.2 (-1.7) | 3744 (-14.9) |
| DAPO | 80.0 (+4.0) | 826 (+76.5) | 85.8 (+6.2) | 3106 (-14.1) | 26.7 (-2.0) | 8583 (-22.3) | 66.3 (+3.0) | 5666 (-25.9) | 46.6 (-0.4) | 5822 (-24.2) | 61.1 (+2.1) | 3822 (-2.0) |
| AdaThink | 82.0 (+6.0) | 772 (+65.0) | 79.6 (+0.0) | 1905 (-47.3) | 23.7 (-5.0) | 7434 (-32.7) | 58.7 (-4.6) | 3983 (-47.9) | 49.8 (+2.8) | 4706 (-38.7) | 58.8 (-0.2) | 2948 (-20.3) |
| Laser-D | 84.9 (+8.9) | 1073 (+129.3) | 85.2 (+5.6) | 2424 (-33.0) | 30.0 (+1.3) | 7271 (-34.2) | 65.8 (+2.5) | 4355 (-43.0) | 53.2 (+6.2) | 4813 (-37.3) | **63.8**(+4.9) | 3221 (-3.6) |
| Laser-DE | 84.1 (+8.1) | 1179 (+151.9) | 84.2 (+4.6) | 2798 (-22.6) | 29.7 (+1.0) | 7960 (-27.9) | 65.2 (+1.9) | 5018 (-34.4) | 50.5 (+3.5) | 5265 (-31.4) | 62.7 (+3.8) | 3604 (+7.1) |
| LCR1 | 75.0 (-1.0) | 443 (-5.3) | 77.6 (-2.0) | 1851 (-48.8) | 19.0 (-9.7) | 7155 (-35.2) | 56.4 (-6.9) | 3897 (-49.0) | 44.0 (-3.0) | 4193 (-45.4) | 54.4 (-4.5) | **2682**(-36.8) |
| JET | 83.8 (+7.8) | 605 (+29.3) | 83.0 (+3.4) | 2072 (-42.7) | 32.0 (+3.3) | 6641 (-39.9) | 66.1 (+2.8) | 3872 (-49.3) | 51.6 (+4.6) | 4121 (-46.3) | 63.3 (+4.4) | 2710 (-29.8) |
| **DeepSeek-R1-Distill-Qwen-7B** | | | | | | | | | | | | |
| Base | 87.0 | 469 | 92.0 | 2918 | 51.3 | 9812 | 78.9 | 6013 | 63.1 | 6782 | 74.5 | 4026 |
| SFT | 87.3 (+0.3) | 438 (-6.6) | 91.4 (-0.6) | 2568 (-12.0) | 48.7 (-2.6) | 9814 (+0.0) | 78.6 (-0.3) | 5836 (-2.9) | 62.7 (-0.4) | 6441 (-5.0) | 73.7 (-0.7) | 3862 (-5.3) |
| DPO | 86.1 (-0.9) | 438 (-6.6) | 90.0 (-2.0) | 2590 (-11.2) | 53.0 (+1.7) | 9552 (-2.6) | 77.2 (-1.7) | 5797 (-3.6) | 60.4 (-2.7) | 6465 (-4.7) | 73.4 (-1.1) | 3839 (-5.8) |
| DAPO | 90.1 (+3.1) | 583 (+24.3) | 91.6 (-0.4) | 2720 (-6.8) | 53.3 (+2.0) | 8414 (-14.2) | 81.7 (+2.8) | 4903 (-18.5) | 63.4 (+0.3) | 5361 (-21.0) | 76.0 (+1.5) | 3405 (-7.2) |
| AdaThink | 88.9 (+1.9) | 304 (-35.2) | 87.8 (-4.2) | 1325 (-54.6) | 50.7 (-0.6) | 8131 (-17.1) | 77.2 (-1.7) | 3871 (-35.6) | 61.3 (-1.8) | 4656 (-31.3) | 73.2 (-1.3) | 2720 (-34.8) |
| Laser-D | 91.6 (+4.6) | 965 (+105.8) | 92.0 (+0.0) | 1950 (-33.2) | 52.7 (+1.4) | 6361 (-35.2) | 82.8 (+3.9) | 3505 (-41.7) | 64.7 (+1.6) | 3755 (-44.6) | 76.8 (+2.3) | 2649 (-9.8) |
| Laser-DE | 91.5 (+4.5) | 948 (+102.1) | 92.4 (+0.4) | 1942 (-33.4) | 53.0 (+1.7) | 5809 (-40.8) | 82.9 (+4.0) | 3357 (-44.2) | 64.6 (+1.5) | 3713 (-45.3) | **76.9** (+2.4) | 2554 (-12.3) |
| LCR1 | 86.0 (-1.0) | 386 (-17.7) | 87.6 (-4.4) | 1313 (-55.0) | 50.0 (-1.3) | 6329 (-35.5) | 76.5 (-2.4) | 3173 (-47.2) | 59.1 (-4.0) | 3575 (-47.3) | 71.8 (-2.6) | **2238** (-40.5) |
| JET | 86.1 (-0.9) | 324 (-30.9) | 91.2 (-0.8) | 2091 (-28.3) | 54.0 (+2.7) | 7981 (-18.7) | 81.0 (+2.1) | 4301 (-28.5) | 63.9 (+0.8) | 5083 (-25.1) | 75.2 (+0.8) | 2999 (-26.3) |

## 4 EXPERIMENTS

### 4.1 EXPERIMENT SETUPS

**Models.** To assess the effectiveness of JET, we adopt two representative and widely used LRMs, Deepseek-R1-Distill-Qwen7B and 1.5B (DeepSeek-AI et al., 2025), as backbone models.

**Datasets.** (i) Training data. We construct a mixed-difficulty training dataset by combining MATH and DAPO-MATH [2]. We then remove all Chinese-language problems, resulting in 14,564 examples. (ii) Test data. We evaluate our model across a diverse suite of benchmarks to assess its performance in both in-domain and out-of-domain scenarios. For in-domain mathematical reasoning, we use datasets of varying difficulty, including AIME 2024[3], MATH500 (Lightman et al., 2023), GSM8K (Cobbe et al., 2021), AMC[4], and Olympiad (He et al., 2024) problems. To measure out-of-domain generalization, we test the model on GPQA-Diamond, CommonsenseQA, and a subset of MMLU created by sampling 2,000 problems per subject. Finally, to ensure statistical robustness, all experiments on AIME 2024 and AMC are independently repeated 10 times.

**Metrics.** We evaluate model performance using the following four metrics: Accuracy (Acc) for correctness, Output Length (Length) for conciseness, Accuracr change $\Delta acc$ to track performance shifts, and the token compression ratio (Eq. 5) to measure efficiency.

**Baselines.** In addition to backbone models, we compare JET with several efficient reasoning methods. Specifically, **(1) Supervised Fine-tuning (SFT)**: Following OVERTHINK (Chen et al., 2024), we construct training datasets from the shortest correct answers generated during our method's rollout stage to fine-tune the backbones; **(2) Direct Preferene Optimization (DPO)**: We also create a preference dataset, by labeling the shortest correct answer as "chosen" and the longest as "reject"; **(3) DAPO**: A widely used RL algorithm for enhancing reasoning capabilities, analogous to R1; **(4) AdaptThink (AdaThink)** (Zhang et al., 2025): An RL algorithm that teaches reasoning models when to think or not think to solve a given query; **(5) IC-R1** (Cheng et al.): A GRPO-based method that improves reasoning efficiency by pruning invalid steps; **(6) Laser** (Liu et al., 2025a): It's also an RL-based method that promotes reasoning efficiency by using a step-function reward based on target sequence length. Detailed experimental settings and hyperparameters for these baselines are provided in the Appendix A.

---

[2]https://huggingface.co/datasets/BytedTsinghua-SIA/DAPO-Math-17k

[3]https://huggingface.co/datasets/math-ai/aime24

[4]https://huggingface.co/datasets/AI-MO/aimo-validation-amc

Table 2: Generalization ability of different methods on other reasoning tasks. Values in parentheses under *ACC* indicate the accuracy change $\Delta acc$ relative to the *Base*. For each benchmark, the value under *Length* reports the mean token compression ratio across all samples in that benchmark. The value under the *AVG* column further averages these per-benchmark compression ratios to summarize the overall compression performance.

| | Methods | CSQA | | GPQA-Diamond | | MMLU | | AVG | |
|---|---|---|---|---|---|---|---|---|---|
| | | ACC | Length | ACC | Length | ACC | Length | ACC | Length |
| DS-Qwen-1.5B | Base | 44.2 | 787 | 32.3 | 5619 | 43.8 | 1306 | 40.1 | 1370 |
| | SFT | 47.0 (+2.8) | 740 (-6.0) | 34.9 (+2.6) | 5288 (-5.9) | 44.7 (+0.1) | 1122 (-14.1) | 42.2 (+2.1) | 1227 (-8.7) |
| | DPO | 44.6 (+0.4) | 690 (-12.3) | 30.3 (-2.0) | 5283 (-6.0) | 44.5 (+0.7) | 1143 (-12.5) | 39.8 (-0.3) | 1221 (-10.3) |
| | DAPO | 45.6 (+1.4) | 479 (-39.1) | 37.9 (+5.6) | 4591 (-18.3) | 46.5 (+1.8) | 856 (-34.4) | 43.3 (+3.2) | 938 (-30.6) |
| | AdaptThink | 48.0 (+3.8) | 887 (+12.7) | 32.3 (+0.0) | 4601 (-18.1) | 44.5 (-2.0) | 1224 (-6.3) | 41.6 (+1.5) | 1299 (-3.9) |
| | Laser-D | 47.1 (+2.9) | 685 (-13.0) | 34.3 (+2.0) | 5352 (-4.8) | 47.2 (+3.4) | 1143 (-12.5) | 42.9 (+2.8) | 1223 (-10.1) |
| | Laser-DE | 47.8 (+3.6) | 685 (-13.0) | 33.8 (+1.5) | 5575 (-0.8) | 46.2 (+2.4) | 1217 (-6.8) | 42.6 (+2.5) | 1279 (-6.9) |
| | LCR1 | 44.3 (+0.1) | 431 (-45.2) | 33.8 (+1.5) | 3678 (-34.5) | 42.7 (-1.1) | 744 (-43.0) | 40.3 (+0.2) | **802** (-40.9) |
| | JET | 45.6 (+1.4) | 407 (-48.3) | 43.4 (+11.1) | 4182 (-25.6) | 44.6 (+0.8) | 715 (-45.3) | **44.5** (+4.4) | 806 (-39.7) |
| DS-Qwen-7B | Base | 63.7 | 631 | 47.5 | 6359 | 60.1 | 1022 | 57.1 | 1191 |
| | SFT | 65.4 (+1.7) | 593 (-6.0) | 45.5 (-2.1) | 5418 (-14.8) | 60.2 (+0.1) | 1044 (+2.2) | 57.0 (-0.1) | 1136 (-6.2) |
| | DPO | 65.9 (+2.2) | 586 (-7.1) | 49.0 (+1.5) | 5519 (-13.2) | 60.1 (+0.0) | 947 (-7.3) | 58.3 (+1.2) | 1083 (-9.2) |
| | DAPO | 65.7 (+2.0) | 519 (-17.7) | 51.0 (+3.5) | 5485 (-13.7) | 58.9 (-1.3) | 907 (-11.3) | 58.5 (+1.4) | 1034 (-14.3) |
| | AdaptThink | 65.9 (+2.2) | 536 (-15.1) | 44.4 (-3.1) | 4820 (-24.2) | 57.3 (-2.8) | 840 (-17.8) | 55.9 (-2.6) | 962 (-19.0) |
| | Laser-D | 65.8 (+2.1) | 600 (-4.9) | 51.0 (+3.5) | 4417 (-30.5) | 62.3 (+2.2) | 1059 (+3.6) | 59.7 (+2.6) | 1090 (-10.6) |
| | Laser-DE | 64.8 (+1.1) | 553 (-12.4) | 46.5 (-1.0) | 3884 (-38.9) | 63.4 (+3.3) | 1027 (+0.5) | 58.2 (+1.1) | 1023 (-16.9) |
| | LCR1 | 65.0 (+1.3) | 404 (-36.0) | 50.0 (+2.5) | 3429 (-46.1) | 56.5 (-3.6) | 611 (-40.2) | 57.2 (+0.1) | **700** (-40.8) |
| | JET | 66.4 (+2.7) | 531 (-15.8) | 52.5 (+5.0) | 5530 (-13.0) | 63.9 (+3.8) | 860 (-15.9) | **60.9** (+3.8) | 1013 (-14.9) |

## 4.2 MAIN RESULTS

**JET achieves substantial output length reduction without compromising accuracy, demonstrating superior efficiency.** Some methods such as LCR1 attain higher compression but at the cost of accuracy. For example, on MATH500 with the 7B model, LCR1 reduces length by over 50% but drops accuracy by 4.4pp, undermining the goal of efficient reasoning. In contrast, JET consistently attains large reductions, averaging 39.7% on the 1.5B model, while maintaining or even improving accuracy across tasks, reflecting a more favorable efficiency and accuracy trade-off.

**JET shows outstanding performance on challenging mathematical reasoning tasks.** On high-level competition datasets such as AIME24 and AMC, JET provides notable gains. With the 7B model, it achieves 54.0 accuracy on AIME24 (+2.7 over Base) and 81.0 on AMC (+2.1 over Base). These improvements indicate that JET efficiently captures critical reasoning steps, reduces redundant computation, and produces higher-quality solutions. Its output is also shorter than other methods, showing a more efficient reasoning process.

**JET also demonstrates a distinct advantage on simpler tasks**. While other methods, such as Laser, achieve comparable performance to JET on some tasks, they tend to produce unnecessarily long reasoning even for easy problems. For instance, on GSM8K, Laser increases token usage by 105.8%, suggesting that its dynamic difficulty-aware mechanism does not adapt effectively to tasks of varying difficulty. In contrast, JET maintains a consistent balance between accuracy and output length across all task difficulties.

**JET maintains stable performance across model scales and families.** We further evaluate JET using DeepSeek-R1-Distill-Llama-8B as the backbone (Table 3). On 8B model, JET improves accuracy by 2 points while reducing token usage by 31% on in-domain tasks. These results demonstrate that JET 's reasoning strategy is robust and generalizes effectively across models of varying sizes.

## 4.3 GENERALIZATION ANALYSIS OF JET

**JET exhibits robust generalization across domains and difficulty levels in commonsense reasoning tasks**, indicating that its effectiveness stems from an optimized, domain-agnostic reasoning framework rather than incidental factors. Although initially developed for mathematical reasoning, JET achieves consistently strong results on CSQA (commonsense judgment), GPQA (professional reasoning), and MMLU (multidisciplinary evaluation), underscoring its versatility.

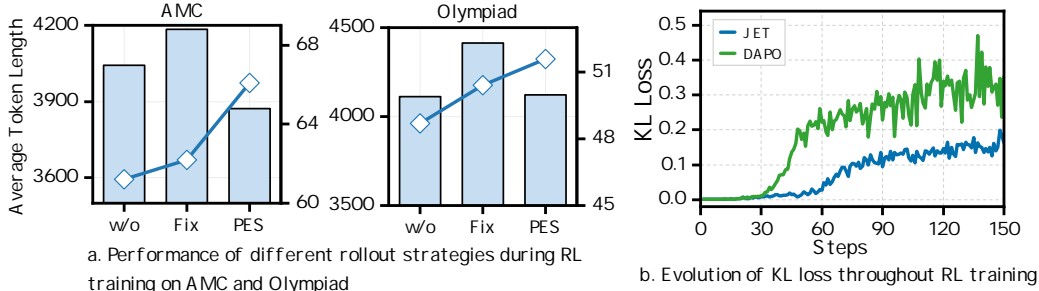

Figure 3: **a.** Performance of different rollout strategies during the RL training. **b.** Evolution of KL loss throughout RL training using the DS-Distill-Qwen-1.5B backbone.

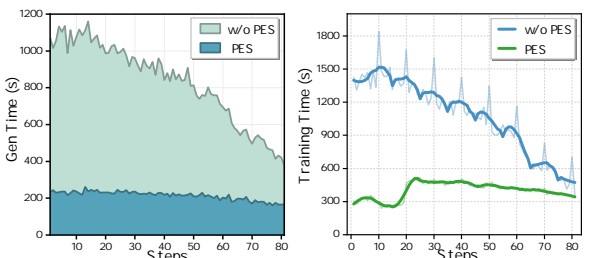

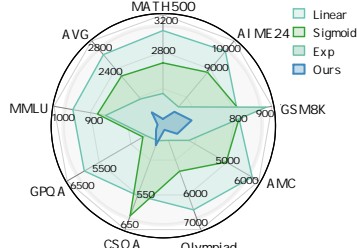

Figure 4: Comparison of rollout generation time and overall RL training time with and without PES. PES accelerates RL training by producing shorter reasoning trajectories.

Figure 5: Average output token length of JET across three length-reward strategies on nine benchmarks.

The largest gains emerge on the challenging GPQA-Diamond benchmark, highlighting **JET 's capacity to handle complex semantic structures**, a central element of its generalization. On this dataset, JET delivers +5.0 improvement with the 7B model, substantially outperforming other methods, and an even larger +11.1 gain with the 1.5B model. The better performance of JET on difficult tasks indicates that JET enhances the model's ability to capture deep reasoning structures, rather than relying on superficial pattern matching, and maintains high performance even in unfamiliar domains.

## 4.4 IMPACT OF PES-INDUCED ANSWER DIVERSITY ON TRAINING

To explore how answer diversity from the PES strategy affects training, we compare PES with fixed-position truncation (Fix) and full reasoning without early stopping (w/o PES). The results are show in Figure 3, and the additional analysis in Appendix C.8. PES improves reasoning accuracy and efficiency by generating paths of varying lengths, exposing the model to different strategies and teaching it when early stopping is beneficial. In contrast, w/o PES suffers from error accumulation in long sequences, and Fix cannot adapt to problem complexity.

## 4.5 DISTRIBUTION CONSISTENCY DURING JET TRAINING

We examine whether JET preserves the model's native probability distribution by analyzing KL-divergence dynamics during training, using DS-Qwen-1.5B as the backbone and comparing with DAPO. As shown in Figure 5, JET consistently maintains a lower KL throughout training, indicating that its truncation strategy does not introduce distributional distortions. This suggests that even when reasoning is truncated, JET preserves the model's sequence-level probability structure, enabling stable reinforcement learning without deviating from the original output distribution.

## 4.6 EFFICIENCY ANALYSIS OF PES

**PES achieves significant computational efficiency.** Instead of performing a costly search for the optimal truncation points, PES employs a simple and effective progressive approach to approximate the optimal reasoning length. By truncating full reasoning trajectories, PES reduces computational overhead during rollout. The shortened trajectories require fewer forward passes and enable faster gradient computation, further accelerating RL training. Compared to the baseline strategy that generates full reasoning chains, PES achieves up to a five-fold speedup in rollout generation and policy optimization, as shown in Figure 4.

## 4.7 Length Reward Design Analysis

We compare our length reward with three alternatives: linear (Linear), exponential decay (Exp), and Sigmoid (Arora & Zanette, 2025a). Linear assigns scores proportionally to reasoning length, Exp penalizes longer outputs more sharply, and Sigmoid provides a smooth transition, moderately penalizing intermediate lengths while strongly discouraging very long reasoning. Detailed formulations are in Appendix C.6, and results are shown in Figure 5. Across eight datasets, the weighted linear reward effectively reduces reasoning length. By encouraging shorter outputs while ensuring a minimum reward, it preserves essential reasoning steps and removes redundancy, achieving a good balance between accuracy and efficiency (Table 4). Further analysis is provided in Appendix C.6.

## 5 Related work

**Enhancing Deep Thinking Capability in LLMs.** Unlike the rapid, heuristic-driven behavior of LLMs, slow-thinking reasoning systems enhance their capabilities by introducing deliberate and analytical reasoning (Snell et al., 2024). These approaches employ test-time scaling, enabling models to allocate more computation and time to reasoning before responding to challenging questions (Muennighoff et al., 2025b). Reinforcement Learning (RL) has become the main technique for building strong reasoning abilities in LLMs during post-training stage. OpenAI's o1 model (Jaech et al., 2024) was the first large-scale use of RL for deep reasoning, showing excellent performance on complex tasks. Soon after, DeepSeek-R1 (DeepSeek-AI et al., 2025) became the first open-source model to match o1's performance, making these techniques available to more researchers. This progress has led to many powerful long reasoning models, including Gemini 2.5 (Comanici et al., 2025), QwQ (Team, 2025), and Phi-4 (Abdin et al., 2024). Recent work has shown that Reinforcement Learning with Verifiable Rewards (RLVR) (Chu et al., 2025; Betley et al., 2025) can greatly improve model's performance on challenge reasoning task, especially in mathematics and programming (Shao et al., 2024; Hu et al., 2025a).

**Efficient Reasoning.** While detailed reasoning often leads to more correct answers, the redundant thought process of LRMs greatly increases the inference time and computational cost, a problem known as "overthinking" (Sui et al., 2025; Feng et al., 2025). Many work have proposed methods to improve reasoning efficiency from different perspectives.
One group of methods sets a fixed token budget for reasoning. They directly control the length of reasoning by setting explicit token limits (Muennighoff et al., 2025a; Sun et al., 2025; Aggarwal & Welleck, 2025; Anthropic, 2025). Examples include CoT-Valve (Ma et al., 2025) and L1 (Aggarwal & Welleck, 2025). However, it is hard to choose the right budget for problems of different difficulty levels. Another line of work teaches the model to adapt its reasoning length to the difficulty of the question. For example, Adar1 (Luo et al., 2025a) and DAST (Shen et al., 2025) build preference datasets to train the model to decide by itself whether to use a "think" or "no-think" mode for each query (Lou et al., 2025; Zhang et al., 2025; Bai et al., 2023). Another growing body of work explores reinforcement learning to achieve efficient reasoning. Methods such as O1-Pruner (Luo et al., 2025b; Qu et al., 2025b; Dai et al., 2025), ThinkPrune (Hou et al., 2025), and Kimi (Team et al., 2025) add length-based penalties to the reward function to encourage concise but accurate reasoning.

## 6 Conclusion

In this paper, we propose Just-Enough Thinking (JET), a method that trains LRMs to proactively terminate unnecessary reasoning and achieve efficient reasoning. JET tackles the difficulty that LRMs seldom produce short reasoning paths during reinforcement learning, leading to biased training samples. Artificially constructed short answers often diverge from the model's natural probability distribution, which hinders effective learning. Inspired by Evidence Accumulation Models, we design a two-stage rollout strategy, where one stage applies trajectory truncation to construct short reasoning paths consistent with the model's natural distribution. This enables the model to observe multiple reasoning paths for the same question that differ in both length and correctness. We also introduce a quality-controlled length reward to guide the model toward more efficient reasoning. Experiments on two representative LRMs demonstrate that JET significantly reduces output length without sacrificing accuracy, and this efficiency generalizes effectively to other reasoning tasks.

## 7 ACKNOWLEDGMENTS

This work was supported by Ant Group.

## 8 ETHICS STATEMENT

This work adheres to the ICLR Code of Ethics. In this study, no human subjects or animal experimentation was involved. All datasets used were sourced in compliance with relevant usage guidelines, ensuring no violation of privacy. We have taken care to avoid any biases or discriminatory outcomes in our research process. No personally identifiable information was used, and no experiments were conducted that could raise privacy or security concerns. We are committed to maintaining transparency and integrity throughout the research process.

## 9 REPRODUCIBILITY STATEMENT

We have made every effort to ensure that the results presented in this paper are reproducible. All code and datasets have been made publicly available in a repository to facilitate replication and verification. The experimental setup, including training steps, model configurations, and hardware details, is described in detail in the paper. Additionally, All datasets are publicly available, ensuring consistent and reproducible evaluation results. We believe these measures will enable other researchers to reproduce our work and further advance the field.

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

APPENDIX

## A EXPERIMENTS DETAILS

**Hardware and Software Platform.** All experiments are conducted on workstations equipped with four NVIDIA A800 PCIe GPUs with 80GB memory each, running Ubuntu 20.04.6 LTS. Our implementation is based on the Verl (Sheng et al., 2025) framework.

**Training Configuration.** We generate rollouts using temperature sampling ($\tau = 1.0$) with enforced end-of-sequence tokens, and employ vLLM for efficient batched decoding with 128 rollout slots and paged attention. During RL training, the maximum prompt length is set to 2,048 tokens, and the maximum response length is 10,000 tokens. Training is conducted for 100 steps with a batch size of 64, and the actor model is optimized using the Adam optimizer with a learning rate of $1 \times 10^{-6}$.

During the rollout stage, we employ a grouped sampling strategy with a group size of $G = 12$ per problem. Specifically, we first generate three complete answers. To obtain partial reasoning trajectories, the initial truncation point is set to $t_0 = 25\%$ of the original length, the increment $\Delta t$ is 25%, and we perform $k = 3$ truncations. At each truncation point, the model is required to produce a final prediction. This process yields nine short answers and three full answers, resulting in a total of 12 responses per query.

For RL optimization, the clipping parameters are set to a low of 0.2 and a high of 0.28. Reward weights are assigned as $w_{\text{acc}} = 0.9$, $w_{\text{f}} = 0.1$, and $w_{\ell} = 1$ to balance accuracy, formatting, and output length during training.

**Inference Configuration.** During inference, we set the temperature to 0.6, the maximum model length to 30,000 tokens, the maximum tokens to 16,000 tokens, and top-p to 0.95.

**Length Reward Design.** In Section 4.7, we introduce two additional length reward strategies: the *linear reward* and the *exponential decay reward*. The linear reward is defined as:

$$r_{\ell}(i) = \begin{cases} \frac{\ell_{\max} - \ell_i}{\ell_{\max} - \ell_{\min} + \varepsilon}, & \text{if } i \in \mathcal{C}, \\ 0, & \text{otherwise.} \end{cases} \tag{13}$$

The sigmoid length reward to provide a smooth penalty for reasoning length (Arora & Zanette, 2025b), which is calculated by:

$$r_{\ell}(i) = \begin{cases} 1 - \alpha \cdot \sigma\left(\frac{\ell_i - \mu_x}{\sigma_x + \varepsilon}\right), & \text{if } i \in \mathcal{C}, \\ 0, & \text{otherwise,} \end{cases} \tag{14}$$

where $\ell_i$ is the token length of the $i$-th response, $\mu_x = \mathbb{E}_{j \in \mathcal{C}}[\ell_j]$ and $\sigma_x = \sqrt{\text{Var}_{j \in \mathcal{C}}(\ell_j)}$ are the mean and standard deviation of lengths over all correct rollouts for $x$, $\sigma(z) = \frac{1}{1+e^{-z}}$ is the sigmoid function, $\alpha \in [0, 1]$ is a hyperparameter, and $\varepsilon > 0$ is a small constant for numerical stability.

The exponential decay reward replaces the linear term with an exponential function, and is defined as

$$r_{\ell}(i) = \begin{cases} \left(\frac{\ell_{\max} - \ell_i}{\ell_{\max} - \ell_{\min} + \varepsilon}\right)^{\alpha} \cdot (1 - \delta) + \delta, & \text{if } i \in \mathcal{C}, \\ 0, & \text{otherwise.} \end{cases} \tag{15}$$

For all experiments, the hyperparameters are fixed as $\alpha = 1.2$, $\delta = 0.05$, and $\varepsilon = 1 \times 10^{-8}$.

**Baseline Implementation.** We provide a detailed description of the baseline implementation.

- For AdaptThink, LCR1, and Laser, we initially attempted to reproduce the results using the official GitHub repositories provided in their papers. However, our reproduced results did not fully match the reported performance. To ensure fairness and avoid weakening the original results, we instead directly download the trained models from Hugging Face and evaluate them with the prompt configurations specified in the papers, while adopting the inference parameters listed in Appendix A.

- For both the SFT and DPO baselines, we sample 12 answers per problem from JET under the *Two-Stage Rollout*. The shortest correct answer serves as the SFT training target and as the preferred response for DPO, while the longest answer is designated as the rejected response for DPO. This process yields training sets of 5.8K samples for DeepSeek-R1-Distill-Qwen-7B and 5.6K samples for the 1.5B model. Both models are fine-tuned with LoRA for one epoch using using a cutoff length of 4,096 tokens, a learning rate of 1.0e-5, and a maximum of 100,000 training samples.

- For DAPO, we employ the same training data and parameter settings as JET. The only modification lies in the rollout and reward configurations, where the length reward is removed and only the accuracy and formatting rewards are retained.

Table 3: Performance of different baselines across various math tasks on DS-Distill-Llama-8B. Values in parentheses under *ACC* indicate the accuracy change $\Delta acc$ relative to the *Base*. For each benchmark, the value under *Length* reports the mean token compression ratio across all samples in that benchmark. The value under the *AVG* column further averages these per-benchmark compression ratios to summarize the overall compression performance.

| Methods | GSM8K | | MATH500 | | AIME24 | | AMC | | Olympiad | | AVG | |
|---|---|---|---|---|---|---|---|---|---|---|---|---|
| | ACC | Length | ACC | Length | ACC | Length | ACC | Length | ACC | Length | ACC | Length |
| | | | | | DeepSeek-Distill-Llama-8B | | | | | | | |
| Base | 88.6 | 1756 | 88.0 | 4299 | 41.0 | 12223 | 76.9 | 7521 | 52.4 | 8193 | 69.4 | 5493 |
| SFT | 69.5 (-19.0) | 410 (-76.7) | 77.6 (-10.4) | 2833 (-34.1) | 34.7 (-6.3) | 9683 (-20.8) | 72.8 (-4.1) | 6392 (-15.0) | 47.3 (-5.2) | 6249 (-23.7) | 60.4 (-9.0) | 3970 (-27.7) |
| DPO | 71.0 (-17.5) | 415 (-76.4) | 78.6 (-9.4) | 3215 (-25.2) | 36.3 (-4.7) | 11314 (-7.4) | 76.4 (-0.5) | 6450 (-14.2) | 49.5 (-3.0) | 6868 (-16.2) | 62.4 (-7.0) | 4288 (-21.9) |
| DAPO | 81.7 (-6.8) | 1027 (-41.5) | 86.6 (-1.4) | 3258 (-24.2) | 40.8 (-0.2) | 9701 (-20.6) | 77.4 (+0.5) | 5463 (-27.4) | 52.4 (0.0) | 6463 (-21.1) | 67.8 (-1.6) | 4081 (-25.7) |
| JET | 91.1 (+2.6) | 1002 (-42.9) | 89.0 (+1.0) | 2922 (-32.0) | 41.0 (0.0) | 9231 (-24.5) | 76.8 (-0.1) | 5261 (-30.1) | 59.1 (+6.7) | 5822 (-28.9) | 71.4 (+2.0) | 3821 (-30.4) |

Table 4: Comparison of accuracy for three reward strategies on eight benchmark datasets.

| Strategy | GSM8K | MATH500 | AIME24 | AMC | Olympiad | CSQA | GPQA | MMLU | AVG |
|---|---|---|---|---|---|---|---|---|---|
| Ours | 83.8 | 83.2 | 32.0 | 66.1 | 51.6 | 45.6 | 43.4 | 44.6 | 56.3 |
| Linear | 82.6 | 82.6 | 31.7 | 64.2 | 50.2 | 45.6 | 38.9 | 45.2 | 55.1 |
| Exp | 85.3 | 84.6 | 33.0 | 63.6 | 52.3 | 46.0 | 43.4 | 45.9 | 56.8 |
| Sigmoid | 81.7 | 84.0 | 30.7 | 66.6 | 46.1 | 45.5 | 38.4 | 46.0 | 54.9 |

Table 5: Effect of accuracy ($w_{acc}$) and length ($w_\ell$) reward coefficients on model performance.

| Methods | GSM8K | | MATH500 | | AIME24 | | AMC | | Olympiad | | AVG | |
|---|---|---|---|---|---|---|---|---|---|---|---|---|
| | ACC | Length | ACC | Length | ACC | Length | ACC | Length | ACC | Length | ACC | Length |
| Base | 76.0 | 468 | 79.6 | 3617 | 28.7 | 11046 | 63.3 | 7644 | 47.0 | 7679 | 58.9 | 4765 |
| $w_\ell = 1.0, w_{acc} = 0.9$ | 83.8 | 605 | 83.2 | 2072 | 32.0 | 6641 | 66.1 | 3872 | 51.6 | 4121 | **63.3** | **2710** |
| $w_\ell = 0.9, w_{acc} = 1.0$ | 82.6 | 726 | 82.2 | 2459 | 28.7 | 7731 | 66.0 | 4783 | 49.3 | 4852 | 61.8 | 3243 |
| $w_\ell = 0.0, w_{acc} = 0.9$ | 84.2 | 886 | 84.0 | 2997 | 29.3 | 9822 | 65.9 | 5671 | 48.7 | 6044 | 62.4 | 3974 |
| $w_\ell = 1.0, w_{acc} = 1.5$ | 81.4 | 985 | 85.4 | 3591 | 30.7 | 10332 | 66.8 | 6727 | 47.7 | 6963 | 62.4 | 4547 |
| $w_\ell = 1.5, w_{acc} = 1.0$ | 79.8 | 981 | 83.4 | 3553 | 25.7 | 10532 | 64.1 | 6406 | 48.6 | 6959 | 60.3 | 4482 |
| $w/incorrect$ | 78.0 | 871 | 82.4 | 3823 | 29.7 | 9711 | 64.2 | 7452 | 46.7 | 6997 | 60.2 | 4658 |

# B  ALGORITHM

Based on the description in Section 3.1, we present the pseudocode of the JET algorithm in Algorithm 1, which outlines its key steps and facilitates the reproducibility of our method.

# C  FURTHER ANALYSIS

## C.1  PERFORMANCE EVOLUTION DURING RL TRAINING

We track how model performance evolves throughout JET training across multiple downstream tasks for both DeepSeek-Distill-Qwen-1.5B and 7B. Figures 6 and 7 illustrate the main trends.

The average token length decreases significantly over training steps, while accuracy remains stable or improves, indicating that our method successfully encourages concise yet accurate responses. In early

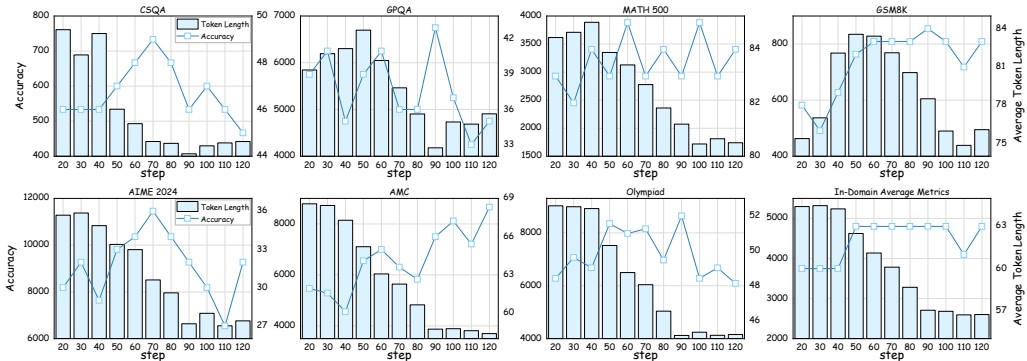

Figure 6: Evolution of accuracy and output length of the DeepSeek-Distill-Qwen-1.5B model across benchmarks during JET training.

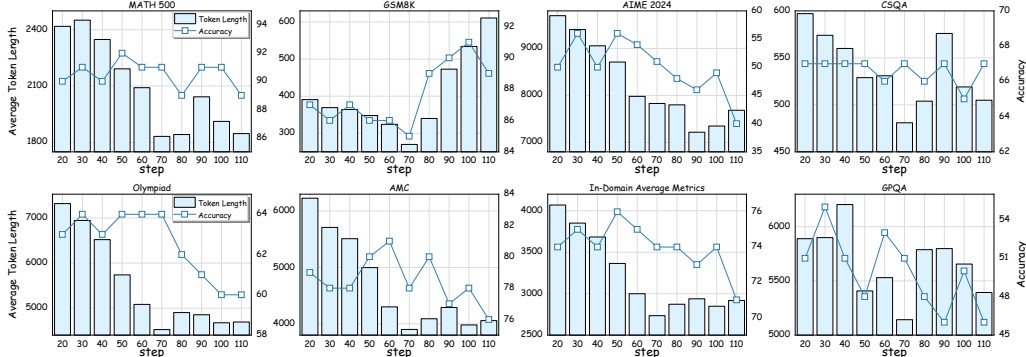

Figure 7: Evolution of accuracy and output length of the DeepSeek-Distill-Qwen-7B model across benchmarks during JET training.

stages, both models exhibit relatively long outputs with moderate accuracy, reflecting an exploration stage where the policy is still learning to balance quality and brevity. As training proceeds, output length drops sharply, especially on structured reasoning tasks such as MATH500, GSM8K, and AMC, where the average token count decreases by more than 50%. Crucially, this compression coincides with sustained or improved accuracy, indicating that shorter outputs are semantically meaningful rather than merely truncated.

The 7B model converges faster and exhibits smoother trends than the 1.5B variant. It achieves higher final accuracy and more consistent length reduction, likely due to its stronger generalization capacity and richer representations. Nonetheless, the smaller model also shows substantial improvement, confirming that the method scales effectively across model sizes.

Overall, the in-domain metrics trace a clear progression from long, low-efficiency outputs to shorter, more precise responses with stable performance. This trajectory highlights the success of our RL design in aligning model behavior with the dual goals of correctness and conciseness.

## C.2 FURTHER ANALYSIS OF JET 'S PERFORMANCE ON OTHER MODELS

To further validate JET 's effectiveness, we evaluate it using DeepSeek-R1-Distill-Llama-8B as the backbone. Since other baselines were not evaluated with this model, we report only a subset of them, and the results ar shown in Table 3.

JET demonstrates strong generalization across backbones, maintaining a stable balance between accuracy and output length. Compared to the base model, it improves accuracy by 2 points while reducing average token usage by 31%. On out-of-domain tasks, JET reduces token usage by 29% while gaining 0.8 points in accuracy, showing that its efficiency and performance advantages extend to larger-scale models.

We also analyze JET on the 1.5B model by plotting accuracy versus average output length in a Pareto-style diagram ( as shown in Figure 8). Methods appearing toward the top-left of the figure

---

**Algorithm 1** Just-Enough Thinking (JET) Algorithm

---

**Require:** Initial policy $\pi_\theta$, training set $\mathcal{Q}$, rollout size $G$, policy updates $\mu$
**Ensure:** Trained policy $\pi_\theta$
1: **for** step $= 1$ to $M$ **do**
2:      Sample mini-batch $\mathcal{Q}_b \subset \mathcal{Q}$
3:      Save current policy: $\pi_{\theta_{\text{old}}} \leftarrow \pi_\theta$
4:      **for** each $q \in \mathcal{Q}_b$ **do**
5:          **Stage 1: Full trajectory rollout**
6:          Generate $G_{\text{full}}$ complete reasoning trajectories. $\{o_i^{\text{full}}\}_{i=1}^{G_{\text{full}}} \sim \pi_{\theta_{\text{old}}}(\cdot \mid q)$
7:          **Stage 2: Truncated trajectory rollout**
8:          **for** each $o_i^{\text{full}}$ and truncation ratio $T \in \mathcal{T}$ **do**
9:              Truncate $o_i^{\text{full}}$ at $T$, append *stop-thinking cue*, and complete: $\hat{o}_{i,T} \sim \pi_{\theta_{\text{old}}}(\cdot \mid q, o_i^{(T)}, z_{\text{stop}})$
10:          **end for**
11:          Collect all trajectories $\mathcal{O}_q$ and compute rewards $r_{\text{acc}}, r_f, r_\ell$ (Eq. 14)
12:          Compute token-level advantages $\hat{A}_{i,t}$ for all trajectories
13:      **end for**
14:      **for** iteration $= 1$ to $\mu$ **do**
15:          Update policy $\pi_\theta$ by maximizing $\mathcal{J}_{\text{DAPO}}$ (Eq. 9)
16:      **end for**
17: **end for**
18: **return** $\pi_\theta$

---

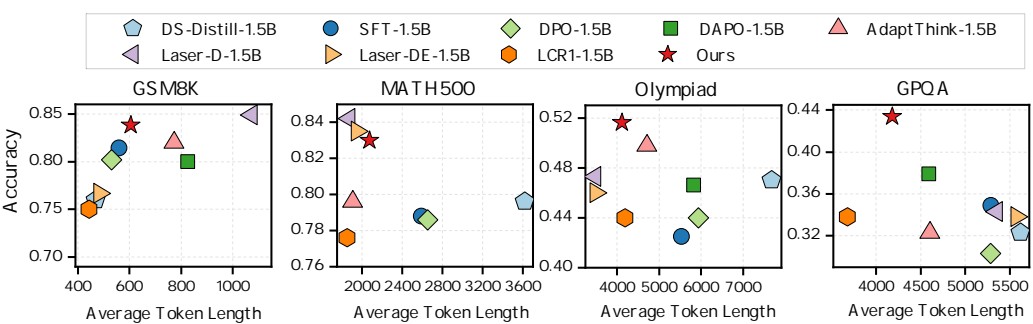

Figure 8: JET demonstrates strong performance across most tasks.

achieve better trade-offs between correctness and efficiency. Across most tasks, JET consistently occupies this favorable region, indicating substantial gains in efficient reasoning compared to other approaches.

## C.3   ANALYSIS OF PROACTIVE STOP-THINKING IN JET

To validate the adaptive reasoning capability of the JET method, we evaluate its performance on mathematical problems of varying difficulty from the MATH500 dataset, and compare it with the manual truncation method (75% cutoff). The results are shown Figure 11.

The JET method can proactively stop reasoning based on task difficulty, significantly reducing token consumption while maintaining high accuracy. This advantage stems from the fact that JET is trained with different reasoning lengths via reinforcement learning, allowing the model to learn the optimal reasoning depth for different tasks. In contrast to manual truncation, which uses a fixed 75% cutoff, the fixed truncation may lead to insufficient reasoning for simpler tasks and overly shallow reasoning for more complex tasks. The ability of JET to dynamically adjust reasoning depth prevents over-reasoning and ineffective reasoning during inference, enabling adaptive adjustments based on task complexity.

## C.4   EFFECT OF ACCURACY AND LENGTH REWARD COEFFICIENTS

JET uses three data formats during training. Since the template only constrains the model to output the final answer in a \boxed{} form and the backbone already shows strong format following. In this section, we do not study the influence of formatting on performance. Instead, this experiment focuses on how the accuracy and length rewards shape model behavior during JET training. We use

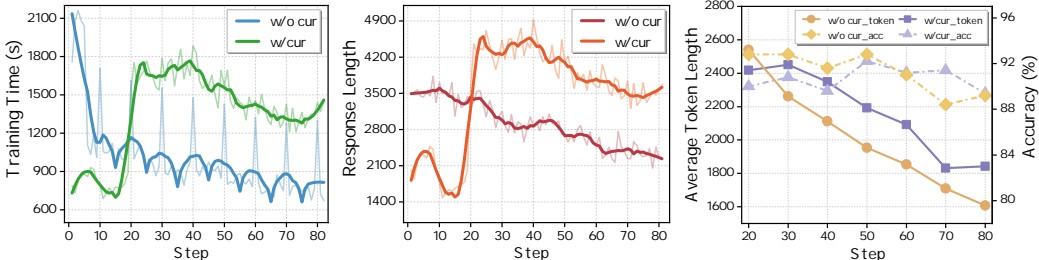

Figure 9: Impact of curriculum learning on JET training. From left to right: RL training time per step, rollout generation time, token output length, and accuracy on MATH500, all measured using the DeepSeek-Distill-Qwen-7B model.

DS-Distill-Qwen-1.5B as the backbone and fixing all other RL settings, we vary the two coefficients and report the results in Table 5.

The findings show that the length reward plays a crucial role in maintaining concise reasoning. Although PES produces short rollouts, it does not change the model's inherent preference; without the length reward, the model quickly drifts toward much longer answers.

Second, we find that the output length becomes excessively long both when the length-reward weight is lower than the accuracy-reward weight and when it is higher. While the resulting trajectory lengths are similar, the underlying causes differ and both reflect forms of reward hacking. Specifically,

• When the length-reward weight is lower than the accuracy-reward weight, the model primarily exploits the accuracy reward. Longer reasoning traces increase the likelihood of achieving a high accuracy score, so the model learns to extend its outputs as a reliable strategy. This represents a standard form of accuracy-driven reward hacking.

• When the length-reward weight is higher than the accuracy-reward weight, the accuracy reward still dominates optimization due to a scale mismatch. It takes discrete values of 0 or 1, producing large and stable gradient signals, whereas the length reward is continuous and has a comparatively smaller range. As a result, the gradient induced by the accuracy reward is much stronger than that from the length reward, even when its explicit weight is smaller. To avoid receiving an accuracy reward of 0, the model favors longer outputs, which again reflects reward-hacking behavior on the accuracy term.

These observations demonstrate that an appropriate balance between accuracy and length rewards is essential for stable training and for preventing both forms of reward hacking.

## C.5 EFFECT OF APPLYING LENGTH REWARD ONLY TO CORRECT ANSWERS

In this section, we investigate whether applying the length reward only to correct answers within each group leads to more effective length reduction and better overall performance. We use the DS-Distill-Qwen1.5B backbone and keep all training configurations identical to the main experiments. We compare two settings: one where the length reward is applied only to correct answers (ours), and another where it is applied to all answers, including incorrect ones (w/incorrect). The results are shown in Table 5.

We observe that Applying the length reward only to correct answers yields clear gains in both accuracy and output length. It achieves higher average accuracy and notably shorter responses compared to the baseline, and consistently outperforms the setting that rewards all answers.

This difference arises because rewarding incorrect answers introduces **conflicting learning signals**. Even with zero accuracy, incorrect answers still receive length reward, causing the model to treat short incorrect and long correct answers as similarly rewarding. This weakens the intended pressure to shorten correct responses and encourages the model to produce longer outputs, a form of reward hacking. Restricting the length reward to correct answers removes this interference. The gradient becomes more focused, giving the model a clear incentive to shorten correct outputs. This leads to more stable learning, more effective length reduction, and improved accuracy. The results demonstrate that applying the length reward only to correct answers is crucial for achieving the desired behavior.

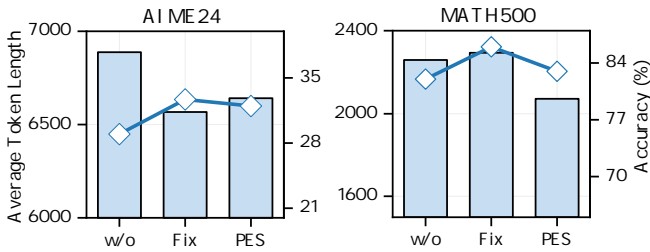

Figure 10: Impact of PES on AIME24 and MATH500.

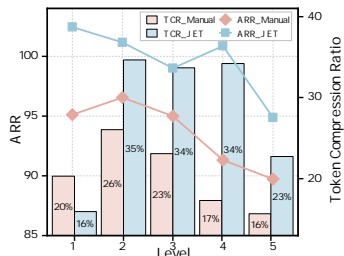

Figure 11: Performance of JET at different difficulty levels.

### C.6 DETAILS OF THREE REWARD STRATEGIES

In the main text, we focus on the impact of the three length reward mechanisms on output length. Here, we also evaluate their effects on accuracy, with the results reported in Table 4.

The exponential weighting reward achieves the highest average accuracy, particularly on challenging math tasks. Our weighted linear reward performs slightly lower on these tasks but remains competitive on knowledge benchmarks such as GPQA and MMLU, demonstrating stability and generalization. The standard linear reward performs worst on most tasks, indicating insufficient incentive for complex problems.

In terms of length control, the linear reward is weak, leaving models prone to noise from verbose reasoning. The exponential decay reward imposes stronger penalties but can cause premature truncation, leading to incomplete reasoning and unstable accuracy on difficult tasks.

### C.7 IMPACT OF CURRICULUM LEARNING ON JET TRAINING

We evaluate the effect of curriculum learning on JET training using the DeepSeek-Distill-Qwen-7B backbone. Figure 9 summarizes the results.

**Training without curriculum learning demonstrates substantially higher efficiency.** Each training step converges in approximately 600–800 seconds, compared to roughly 1,400 seconds per step with curriculum learning, representing nearly a 50% reduction in computational overhead.

In terms of model outputs, **the non-curriculum strategy produces more concise responses without compromising correctness.** Both methods achieve comparable accuracy on the MATH500 dataset, whereas curriculum learning tends to generate longer answers relatively.

These observations can be explained as follows. Curriculum learning is designed to gradually expose the model to increasingly difficult samples, which can help in scenarios with high variability in sample difficulty or when the model is prone to local optima. However, in our JET setting, the training data and rollout strategy already provide diverse and informative samples across difficulty levels. As a result, random sampling without curriculum learning sufficiently exposes the model to the necessary learning signals, allowing it to achieve similar or better performance with reduced training cost. Moreover, the direct exposure to diverse samples enables the model to learn to produce concise yet correct answers more effectively.

### C.8 FURTHER ANALYSIS OF PES

In Section 4.4, we analyze the impact of exposing the model to diverse reasoning paths during training. Here, we also include results on AIME24 and MATH500, shown in Figure 10, and further analyze the effect of PES.

**Longer reasoning sequences do not always lead to better performance; they can introduce error propagation and noise that degrade final accuracy.** Comparing PES with the no-PES baseline reveals a counterintuitive finding: while generating complete reasoning chains might seem advantageous, it actually underperforms PES. Across datasets of varying difficulty, PES consistently surpasses all baseline strategies.

C.9 CASE STUDIES

We present two types of cases: (i) comparisons of JET with the Base and AdaptThink models in answering questions (Cases 1), and (ii) examples where the trained JET method actively terminates its reasoning process when sufficient information has been accumulated (Cases 2 and 3).

**Case 1:** Comparison with Base and AdaptThink models. In this case, the Base 7B model produces a correct answer but generates an overly long response, consuming 4,087 tokens. AdaptThink produces a much shorter answer, using only 636 tokens, but the response is incorrect. In contrast, JET produces a concise and correct answer with just 1,537 tokens. This demonstrates that JET *effectively balances accuracy and brevity, generating answers that are both correct and substantially shorter than the baseline outputs.*

**Case 2:** Active early stopping based on confidence. During reasoning, JET detects that sufficient information has been accumulated and that its answer is likely correct. The model then terminates the reasoning process early and outputs the final answer. This highlights JET *'s ability to adaptively determine when to stop reasoning, reducing unnecessary computation while maintaining correctness.*

**Case 3:** Early termination upon reaching the correct answer. In this case, JET identifies during reasoning that it has already obtained the correct solution. The model then terminates the reasoning process immediately and outputs the final answer. This demonstrates JET 's ability to recognize when further reasoning is unnecessary, effectively reducing redundant computation while ensuring correctness.

Overall, these cases show that JET can simultaneously achieve concise, correct, and well-formatted outputs, and that its adaptive reasoning strategy allows the model to terminate reasoning when sufficient information is obtained. Compared to baseline methods, JET improves efficiency, output quality, and alignment with task-specific requirements.

---

**Case Study 1: JET VS Base / AdaptThink**

**/* Question */**

What integer $n$ satisfies $0 \leq n < 18$ and $n \equiv -11213141 \pmod{18}$?

**/* Base Response (Total Tokens: 4087)*/**

**<think>**

Okay, so I have this problem where I need to find an integer $n$ such that $0 \leq n < 18$ and $n \equiv -11213141 \pmod{18}$.

Hmm, okay, so I need to find the equivalent positive residue of $-11213141$ modulo 18. That means I need to figure out what $-11213141$ is congruent to modulo 18, and then express it as a number between 0 and 17 inclusive.

First, I remember that modulo operation with a negative number can sometimes be tricky, so maybe I should start by finding the positive equivalent of $-11213141$ modulo 18.

Alternatively, I can find the remainder when $-11213141$ is divided by 18. But since negative numbers can complicate things, perhaps it's easier to find the modulo of the absolute value and then adjust the sign accordingly.

*( ... more reasoning tokens omitted for brevity )*

Therefore, $11213141 \equiv 5 \pmod{18}$, so $-11213141 \equiv -5 \pmod{18}$, which is 13 $\pmod{18}$.

So, yes, $n = 13$.

I think that's solid. So, the integer $n$ is 13.

**</think>**

To find the integer $n$ such that $0 \leq n < 18$ and $n \equiv -11213141 \pmod{18}$, we need to determine the equivalent positive residue of $-11213141$ modulo 18.

First, we compute $11213141 \mod 18$. We use the property that $10^k \equiv 10 \pmod{18}$ for $k \geq 1$ and $10^0 \equiv 1 \pmod{18}$.

Breaking down 11213141 digit by digit and computing each term modulo 18:

- Position 0: $1 \times 1 = 1$ - Position 1: $4 \times 10 = 40$ - Position 2: $1 \times 10 = 10$ - Position 3: $3 \times 10 = 30$ - Position 4: $1 \times 10 = 10$ - Position 5: $2 \times 10 = 20$ - Position 6: $1 \times 10 = 10$ - Position 7: $1 \times 10 = 10$

Summing these results: $1 + 40 + 10 + 30 + 10 + 20 + 10 + 10 = 131$

Next, we find $131 \mod 18$: $131 \div 18 = 7$ remainder 5

Thus, $131 \equiv 5 \pmod{18}$.

Therefore, $11213141 \equiv 5 \pmod{18}$. Hence, $-11213141 \equiv -5 \pmod{18}$.

To convert $-5$ to a positive residue modulo 18: $-5 + 18 = 13$

Thus, the integer $n$ that satisfies the given condition is $\boxed{13}$. ✓

**/\* AdaptThink Response (Total Tokens: 636)\*/**
**<think></think>**

To find the integer $n$ such that $0 \leq n < 18$ and $n \equiv -11213141 \pmod{18}$, we need to determine the remainder when $-11213141$ is divided by 18.

First, we can simplify the problem by finding the remainder when 11213141 is divided by 18, and then take the negative of that remainder, adjusting it to be within the range 0 to 17 if necessary.

1. **Find the remainder of** 11213141 **divided by 18:**

- We can use the property that the remainder of a number modulo 18 is the same as the sum of its digits modulo 18. This is because 18 is a multiple of 9, and the sum of the digits of a number is congruent to the number modulo 9. However, for modulo 18, we need to consider the alternating sum of digits in groups of two from the right.

2. **Calculate the alternating sum of digits in groups of two:**

- Let's break down 11213141 into groups of two from the right: $11, 21, 31, 41$. - Now, compute the alternating sum: $11 - 21 + 31 - 41$.

- Calculate step-by-step: - $11 - 21 = -10$ - $-10 + 31 = 21$ - $21 - 41 = -20$

3. **Find the remainder of** $-20$ **modulo 18:**

- $-20 \div 18 = -2$ with a remainder. To find the positive remainder, add 18 to $-20$: - $-20 + 18 = -2$ (still negative, so add another 18)

- $-2 + 18 = 16$

- Therefore, the remainder is 16.

4. **Determine the final result:**

- Since we initially had $-11213141$, the remainder when divided by 18 is the negative of the above remainder, adjusted to be within 0 to 17.

- $-16 \mod 18 = 2$ because $-16 + 18 = 2$.

Thus, the integer $n$ that satisfies $0 \leq n < 18$ and $n \equiv -11213141 \pmod{18}$ is:

$\boxed{2}$ ✗

**/\* JET Response (Total Tokens: 1537)\*/**

**<think>** To find the integer $n$ such that $0 \leq n < 18$ and $n \equiv -11213141 \pmod{18}$, I'll start by simplifying the problem.

First, I recognize that finding $n$ modulo 18 is equivalent to finding the remainder when $-11213141$ is divided by 18.

Instead of dealing with the negative number directly, I can find the remainder of 11213141 when divided by 18 and then subtract that from 18 to get the equivalent positive remainder.

To find the remainder of 11213141 divided by 18, I'll use the property that a number is congruent to the sum of its digits modulo 9. Since 18 is a multiple of 9, this method is applicable.

Calculating the sum of the digits of 11213141:

1 + 1 + 2 + 1 + 3 + 1 + 4 + 1 = 14.

Now, 14 modulo 9 is 5, which means $11213141 \equiv 5 \pmod{9}$.

Since 18 is twice 9, I need to find a number that is 5 modulo 9 but also adjusts for the fact that we're dealing with modulo 18.

Considering $-11213141 \equiv -5 \pmod{9}$, which simplifies to $4 \pmod{9}$ because $-5 + 9 = 4$.

To find the corresponding modulo 18, I can use the fact that 4 and 18 share the same remainder when divided by 9. Thus, $-11213141 \equiv 4 \pmod{18}$.

Therefore, the integer $n$ that satisfies the given condition is 4. **</think>** To find the integer $n$ satisfying $0 \leq n < 18$ and $n \equiv -11213141 \pmod{18}$, we can compute $-11213141$ modulo 18.

First, note that $-11213141$ modulo 18 is equivalent to finding the remainder when $-11213141$ is divided by 18. Since negative remainders can be converted to positive by adding 18, we can calculate:

$$-11213141 \mod 18 = (18 - (11213141 \mod 18)) \mod 18$$

Calculating $11213141 \mod 18$, we find that 11213141 leaves a remainder of 5 when divided by 9 (since the sum of its digits is 14, and $14 \mod 9 = 5$).

*( ... more reasoning tokens omitted for brevity )*

Therefore, the correct final answer is $\boxed{13}$. ✓

---

## Case Study 2: Active Stop-Thinking of JET

**/\* Question \*/**

Convert the point $(0, 3)$ in rectangular coordinates to polar coordinates. Enter your answer in the form $(r, \theta)$, where $r > 0$ and $0 \leq \theta < 2\pi$.

**/\* JET Response \*/**

**<think>**

Okay, so I need to convert the rectangular coordinate $(0, 3)$ to polar coordinates. Hmm, I

remember that polar coordinates are represented as $(r, \theta)$, where $r$ is the radius and $\theta$ is the angle.

First, let me recall the formulas for conversion. I think $r$ is calculated using the distance formula, which is the square root of $(x^2 + y^2)$. So,

$r = \sqrt{x^2 + y^2}$.

In this case, $x = 0$ and $y = 3$. Plugging those in,

$r = \sqrt{0^2 + 3^2} = \sqrt{0 + 9} = \sqrt{9} = 3$.

So, $r = 3$. That part was straightforward.

Now, I need to find $\theta$, the angle. I remember that $\theta$ can be found using the arctangent function since

$\tan \theta = \frac{y}{x}$.

But wait, in this case, $x = 0$. Hmm, dividing by zero might be an issue here. Let me think about it. When $x = 0$, the point lies on the $y$-axis. Since $y = 3 > 0$, the point is on the positive $y$-axis, which should correspond to an angle of $\frac{\pi}{2}$ radians.

But just to make sure, let me visualize the point $(0, 3)$. On the rectangular coordinate system, that's located straight up along the $y$-axis. So, from the positive $x$-axis, the angle to reach there is $90°$, which is $\frac{\pi}{2}$ radians.

So, $\theta = \frac{\pi}{2}$.

Putting it all together, the polar coordinates are $\left(3, \frac{\pi}{2}\right)$. Let me just double-check if $r$ is positive and $\theta$ is within the required range. $r = 3 > 0$, and $\frac{\pi}{2} \in [0, 2\pi)$, so that's correct.

**I think that's it. I don't see any mistakes here. Yeah, I'm confident with this answer.**

**</think>**

The polar coordinates are $\boxed{\left(3, \dfrac{\pi}{2}\right)}$.

---

## Case Study 3: Active Stop-Thinking of JET

**/* Question */**

What is the smallest positive perfect cube that can be written as the sum of three consecutive integers?

**/* JET Response */**

**<think>**

Okay, so I need to find the smallest positive perfect cube that can be written as the sum of three consecutive integers. Hmm, let's see. I remember that consecutive integers are numbers that come one after the other without any gaps. For example, $1, 2, 3$ or $5, 6, 7$. So, the question is asking for a perfect cube (like $1, 8, 27$, etc.) that can be expressed as the sum of such a triplet.

Let me start by defining the three consecutive integers. Let's call the first integer $n$. Then the next two integers would be $n + 1$ and $n + 2$. So, the sum of these three numbers would be $n + (n + 1) + (n + 2)$.

Let me compute that:

$n + n + 1 + n + 2 = 3n + 3.$

So, the sum simplifies to $3n + 3$. I can factor that as $3(n + 1)$. So, the sum is three times one more than the first integer in the triplet.

Now, the sum needs to be a perfect cube. Let me denote the perfect cube as $k^3$, where $k$ is a positive integer. Therefore, I have the equation:

$3(n + 1) = k^3.$

From this equation, I can express $n + 1$ as $\frac{k^3}{3}$. Since $n + 1$ has to be an integer (as $n$ is an integer), $k^3$ must be divisible by 3. That means $k$ itself must be a multiple of 3, because if a prime number divides a cube, it must divide the base. So, let me let $k = 3m$, where $m$ is a positive integer.

Substituting back, $k = 3m$, so $k^3 = (3m)^3 = 27m^3$. Plugging this into the equation:

$3(n + 1) = 27m^3.$

Dividing both sides by 3:

$n + 1 = 9m^3.$

Therefore, $n = 9m^3 - 1$. Since $n$ is the first integer in the triplet, it just needs to be an integer, which it will be as long as $m$ is an integer.

Now, the sum $3(n+1) = 27m^3$ must be a positive perfect cube. The smallest positive perfect cube is 1, but let's see if that works. If $k^3 = 1$, then $3(n + 1) = 1$, which would imply $n + 1 = \frac{1}{3}$, so $n = -\frac{2}{3}$. But $n$ has to be an integer, so that doesn't work.

Next, try $k^3 = 8$, which is $2^3$. Then, $3(n + 1) = 8$ implies $n + 1 = \frac{8}{3}$, so $n = \frac{5}{3}$. Again, not an integer.

Moving on to the next cube, $k^3 = 27$. Then, $3(n + 1) = 27$ gives $n + 1 = 9$, so $n = 8$. That's an integer! Perfect. So, let's check if the sum works. The triplet would be $8, 9, 10$. Adding them up: $8 + 9 + 10 = 27$, which is indeed $3^3$.

Wait a second, let me confirm if there's a smaller cube than 27 that satisfies the condition. The cubes are $1, 8, 27, 64$, etc. We saw that 1 and 8 don't work because they don't result in integer starting points for the triplet. So, 27 is the next one, and it works. So, 27 is the smallest such cube.

Therefore, the final answer is 27. **I can just write that in a box.**

**</think>**

The smallest positive perfect cube that can be written as the sum of three consecutive integers is $\boxed{27}$.

## D    LLM USAGE

Large Language Models (LLMs) were used to aid in the writing and polishing of the manuscript. Specifically, we used an LLM to assist in refining the language, improving readability, and ensuring clarity in various sections of the paper. The model helped with tasks such as sentence rephrasing, grammar checking, and enhancing the overall flow of the text.

It is important to note that the LLM was not involved in the ideation, research methodology, or experimental design. All research concepts, ideas, and analyses were developed and conducted by the authors. The contributions of the LLM were solely focused on improving the linguistic quality of the paper, with no involvement in the scientific content or data analysis.

The authors take full responsibility for the content of the manuscript, including any text generated or polished by the LLM. We have ensured that the LLM-generated text adheres to ethical guidelines and does not contribute to plagiarism or scientific misconduct.

