# OpenReview forum: "Your Models Have Thought Enough: Training Large Reasoning Models to Stop Overthinking"
_ICLR.cc/2026/Conference — ICLR 2026 Poster_

### Official Review · Reviewer_tagM · 2025-10-26

**Soundness:** 3
**Presentation:** 2
**Contribution:** 2
**Rating:** 4
**Confidence:** 4

**Summary:**

Just-Enough Thinking (JET) tries to address the well-known **overthinking** problem in Large Reasoning Models (LRMs) through introducing (1) a trajectory truncation mechanism that helps early terminate unnecessary reasoning and increases effective rollout sizes, and (2) a quality-controlled length reward used in their DAPO-based RL policy optimization. Through experiments on `DeepSeek-Distill-Qwen` models (training through math datasets), the paper demonstrates considerable reduction in reasoning length on both in-domain and out-of-domain datasets without compromising accuracies.

**Strengths:**

- Overall the paper's writing is clear and easy-to-follow.
- The piloting experiment provides a good intuition and justification on the huge compressibility within LRM's reasoning traces (given that the full reasoning trace leads to correct answer in the first place).
- The main experiments seem comprehensive, including a range of in-domain and out-of-domain tasks and multiple alternative efficient reasoning baselines.
- The paper also includes concrete case studies in the appendix.

**Weaknesses:**

The weaknesses below are **ranked from high priority to low**

_(priority in terms of how they would influence my final rating to this manuscript):_

**Weakness 1: Limited evaluation beyond `Deepseek-Distilled-Qwen`**

Recent RLVR research (not limited to efficient reasoning) often relies on the `Qwen` + `Math` setup. However, several studies -- such as *“RL with One Example”* [[1]](https://arxiv.org/abs/2504.20571) and *“RL with Random Rewards”* [[2]](https://www.interconnects.ai/p/reinforcement-learning-with-random) -- have raised concerns that performance gains might stem from factors like mid-training on math problems or potential data leakage, rather than the RL signal itself. While I am not asserting that such confounding effects necessarily apply here, it would strengthen the paper’s robustness to demonstrate that the reported improvements generalize to other model families beyond Qwen (e.g., the Llama series).

**Weakness 2: Incomplete treatment of overthinking phenomena**

While the paper’s pilot experiment provides an interesting motivation for truncation -- showing that correct reasoning traces are often compressible -- it only captures one side of the broader “overthinking” issue in LLM reasoning. The analysis (particularly the `ARR` metric) is conditioned on problems that are *already solved correctly* with full reasoning traces. This design primarily reflects case (i): verbose reasoning where the correct answer appears early but is followed by unnecessary continuation.

However, overthinking also includes case (ii): overly long or convoluted reasoning that ultimately leads to *incorrect* answers due to confusion or drift. The current truncation approach and pilot study seem to overlook this dimension. Similarly, in the reward design, the use of a constant **0** length reward for all incorrect rollouts fails to distinguish between long and short reasoning traces that are both incorrect.

**Weakness 3: Missing ablations and unclear design justifications**

Although the paper includes numerous additional experiments in the appendix, several (e.g., the curriculum learning experiment) appear unrelated to the main narrative and do not directly clarify the proposed method’s key design choices. More targeted experiments are needed, especially in two areas:

1. **Ablation: Effect of PES without the length reward.**
   It remains unclear how much of the observed improvement comes from the proposed length reward versus PES. Since PES-based truncation effectively constructs “low-likelihood” reasoning trajectories, GRPO/DAPO training might already encourage shorter reasoning paths by increasing the likelihood of these constructed rollouts. An ablation removing the length reward would help isolate its true contribution.

2. **Design Choice: Comparison with established length reward baselines.**
   While the paper proposes alternative reward designs and demonstrates why the chosen one performs better, prior works such as [*Training Efficient*](https://arxiv.org/abs/2502.04463) and [*ShorterBetter*](https://arxiv.org/abs/2504.21370) have already explored efficient reasoning using length-based rewards. Comparing directly against these established formulations -- rather than self-introduced variants -- would provide a more meaningful and standardized evaluation of the proposed approach.

**Questions:**

In addition to questions mentioned in the `Weaknesses` section:

1. What's the rollout size for baseline methods like DAPO/AdaThink?

2. Why finetune SFT/DPO with LoRA in baseline implementation (instead of full finetuning to be consistent with the RL training)?

3. It seems that some important citations (esp. for the baseline methods) are missing, e.g. the Laser paper.

---

> ### Author Response · Authors · 2025-11-24
> **Response to Reviewer tagM(1/3)**
>
> Thank you for your valuable feedback to help us improve our paper. We have revised our paper based on your feedback. We detail our response below and please kindly let us know if our response addresses your concerns.
>
> > **Q1:** Limited evaluation beyond Deepseek-Distilled-Qwen
>
> **A1:** We thank the reviewer for the suggestion. The two backbones we initially selected are commonly used in prior work on efficient reasoning. To further validate JET’s performance, we added DeepSeek-R1-Distill-Llama-8B as a backbone, with results shown in Table 1. Since the referenced baselines did not use this backbone, we report only a subset of them.
>
> We find that **JET demonstrates strong generalization across backbones**, maintaining a stable balance between accuracy and output length. Quantitatively, compared to the base model, JET improves accuracy by 2 points while reducing average token usage by 31%. On out-of-domain tasks, it reduces token usage by 29% while gaining 0.8 points in accuracy. Detailed experimental results and further analyses are provided in the appendix C.2.
>
> **Table 1. The comparison performance on DeepSeek-Distill-Llama-8B.**
> | Methods | GSM8K |  | MATH500 |  | AIME24 |  | AMC |  | Olympiad |  | AVG |  |
> | :--- | :--- | :--- | :--- | :--- | :--- | :--- | :--- | :--- | :--- | :--- | :--- | :--- |
> |  | ACC | Length | ACC | Length | ACC | Length | ACC | Length | ACC | Length | ACC | Length |
> | Base | 88.6 | 1756 | 88.0 | 4299 | 41.0 | 12223 | 76.9 | 7521 | 52.4 | 8193 | 69.4 | 5493 |
> | SFT | 69.5 | 410 | 77.6 | 2833 | 34.7 | 9683 | 72.8 | 6392 | 47.3 | 6249 | 60.4 | 3970 |
> | DPO | 71.0 | 415 | 78.6 | 3215 | 36.3 | 11314 | 76.4 | 6450 | 49.5 | 6868 | 62.4 | 4288 |
> | DAPO | 81.7 | 1027 | 86.6 | 3258 | 40.8 | 9701 | 77.4 | 5463 | 52.4 | 6463 | 67.8 | 4081 |
> | JET | 91.1 | 1002 | 89.0 | 2922 | 41.0 | 9231 | 76.8 | 5261 | 59.1 | 5822 | **71.4** | **3821** |
>
> >**Q2:** The pilot experiment only examines truncation on reasoning traces that ultimately produce correct answers, thereby overlooking cases of “overthinking” that lead to incorrect answers.  The current truncation approach and pilot study seem to overlook this dimension.
>
> **A2:** Thank you for the helpful suggestion. We clarify the role of truncation both in pilot experiments and in JET training, highlighting its effect on mitigating “overthinking” that leads to incorrect answers.
>
> 1. **Truncation in pilot experiments:** In the pilot study, we initially examined truncation only on reasoning traces that ultimately produced correct answers. We observed that (i) the model often gathers sufficient information early in the reasoning process, so truncating the trajectory can preserve accuracy, and (ii) truncated reasoning traces remain consistent with the model’s probability distribution.
> Following your suggestion, we **extended our analysis to truncation on reasoning traces that originally led to incorrect answers**, in order to further support the conclusions drawn from the pilot study. We define the Error Correction Rate(ECR) as the proportion of initially incorrect answers that become correct after truncation. Under the same settings, we observed:
>
> - Retaining 75% of the trajectory: ECR = 10.0%, average token length reduced from 8985 to 8040
>
> - Retaining 50% of the trajectory: ECR = 12.5%, average token length reduced from 8985 to 4668
>
> - Retaining 25% of the trajectory: ECR = 22.5%, average token length reduced from 8985 to 2686
>
> 2. **Truncation in JET training:** During JET training, **truncation is applied to both correct and incorrect trajectories.** Training follows a two-stage rollout: the first stage generates the full reasoning trajectory, and the second stage truncates it at various points (Section 3.1). Even if the full trajectory is incorrect, truncation is still performed, which encourages the model to produce concise reasoning while maintaining accuracy.
>
> Overall, these analyses demonstrate that truncation not only preserves correct answers but also mitigates errors from overthinking, supporting the effectiveness of JET’s design. The corresponding discussion and results have been added to the updated manuscript.
>
> >**Q3:** Similarly, the reward function assigns a constant 0 length reward to all incorrect rollouts, failing to distinguish between long and short incorrect reasoning traces.
>
> **A3:** We adopt this design for two reasons. First, rewarding short but incorrect reasoning traces could lead to severe reward hacking: the model might intentionally produce short incorrect answers to obtain a reward comparable to that of long but correct trajectories, which would compromise accuracy. Second, if the answer is already incorrect, its length is meaningless. Distinguishing long from short incorrect trajectories does not add any useful information.

---

> ### Author Response · Authors · 2025-11-24
> **Response to Reviewer tagM(2/3)**
>
> > **Q4:** It is unclear how much improvement comes from the length reward versus PES. Since PES truncation already encourages shorter reasoning, an ablation without the length reward would clarify its actual contribution.
>
> **A4:** Thank you for the helpful suggestion. To evaluate the contribution of the length reward, we conducted an ablation study using the 1.5B backbone under the same training setup. Specifically, we compared JET with the length reward (w_l=1, w_acc=0.9) and without it (w_l=0, w_acc=0.9), and further analyzed how varying the length-reward and accuracy coefficients affects overall performance. The results are shown in Table 2.
>
> Our experiments indicate that the length reward is essential for encouraging concise reasoning. While PES generates shorter trajectories during the rollout stage, it does not modify the model’s underlying preference; without the length reward, the model tends to generate substantially longer trajectories. The detailed discussion and additional analysis of this part have been included in the Section 4.7 and Appendix C.4.
>
> **Table2: Effect of accuracy $w_{acc}$ and length $w_{\ell}$ reward coefficients on model performance.**
> | Methods | GSM8K |  | MATH500 |  | AIME24 |  | AMC |  | Olympiad |  | AVG |  |
> | :--- | :--- | :--- | :--- | :--- | :--- | :--- | :--- | :--- | :--- | :--- | :--- | :--- |
> |  | ACC | Length | ACC | Length | ACC | Length | ACC | Length | ACC | Length | ACC | Length |
> | Base | 76.0 | 468 | 79.6 | 3617 | 28.7 | 11046 | 63.3 | 7644 | 47.0 | 7679 | 58.9 | 4765 |
> | $w_{\ell} = 1.0, w_{acc} = 0.9$ | 83.8 | 605 | 83.2 | 2072 | 32.0 | 6641 | 66.1 | 3872 | 51.6 | 4121 | **63.3** | **2710** |
> | $w_{\ell} = 0.9, w_{acc} = 1.0$ | 82.6 | 726 | 82.2 | 2459 | 28.7 | 7731 | 66.0 | 4783 | 49.3 | 4852 | 61.8 | 3243 |
> | $w_{\ell} = 0.0, w_{acc} = 0.9$ | 84.2 | 886 | 84.0 | 2997 | 29.3 | 9822 | 65.9 | 5671 | 48.7 | 6044 | 62.4 | 3974 |
> | $w_{\ell} = 1.0, w_{acc} = 1.5$ | 81.4 | 985 | 85.4 | 3591 | 30.7 | 10332 | 66.8 | 6727 | 47.7 | 6963 | 62.4 | 4547 |
> | $w_{\ell} = 1.5, w_{acc} = 1.0$ | 79.8 | 981 | 83.4 | 3553 | 25.7 | 10532 | 64.1 | 6406 | 48.6 | 6959 | 60.3 | 4482 |

---

> ### Author Response · Authors · 2025-11-24
> **Response to Reviewer tagM(3/3)**
>
> > **Q5:** The paper should compare its reward design with established length-based reward methods from prior work, rather than only with self-introduced variants, to provide a more meaningful evaluation.
>
> **A5:**  We thank the reviewer for the suggestion. To provide a more meaningful evaluation, we compare our reward design with the length-based reward proposed in [1]. We use the 1.5B backbone model and keep all training parameters and data consistent; only the reward function differs. The method in [1] employs a sigmoid-based scheme to assign rewards based on trajectory length.
>
> Our experimental results, as shown in Table 3, across multiple datasets are as follows: our weighted length reward mechanism achieves an average accuracy of 56.3% with an average token length of 1,786, whereas the length reward in [1] achieves 54.9% accuracy with an average token length of 2,300.
>
> These results demonstrate that our weighted length reward effectively discriminates trajectories based on their lengths and achieves a superior trade-off between accuracy and reasoning length. In particular, it produces more concise reasoning while maintaining higher accuracy.
>
> **Table 3. The effect of different length reward mechnism on JET across eight dataset.**
> |  | GSM8K |  | MATH500 |  | AIME24 |  | AMC |  | Olympiad |  | CSQA |  | GPQA |  | MMLU |  | AVG |  |
> | :--- | :--- | :--- | :--- | :--- | :--- | :--- | :--- | :--- | :--- | :--- | :--- | :--- | :--- | :--- | :--- | :--- | :--- | :--- |
> |  | ACC | Length | ACC | Length | ACC | AVG_Length | ACC | Length | ACC | Length | ACC | Length | ACC | Length | ACC | Length | ACC | Length |
> | ours | 83.8 | 605 | 83.0 | 2072 | 32.0 | 6641 | 66.1 | 3872 | 51.6 | 4121 | 45.6 | 407 | 43.4 | 4182 | 44.6 | 715 | **56.3** | **1786** |
> | Sigmoid[1] | 81.7 | 787.0 | 84.0 | 2680.0 | 30.7 | 8513.0 | 66.6 | 5148.0 | 46.1 | 5319.0 | 45.5 | 611 | 38.4 | 4519 | 46.0 | 880 | 54.9 | 2300 |
>
> > **Q6:** What's the rollout size for baseline methods like DAPO/AdaThink?
>
> **A6:**  We thank the reviewer for the question. All RL-based baseline methods, were trained using the same hyperparameters for a fair comparison. In particular, the rollout size is set to 12. Additional details on the training parameters can be found in the Training Configuration section of the paper.
>
> > **Q7:** Why finetune SFT/DPO with LoRA in baseline implementation (instead of full finetuning to be consistent with RL training)?
>
> **A7:** Actually, we also experimented with full fine-tuning. However, since our backbone is a large reasoning model that tends to generate excessively long outputs, full fine-tuning on the synthetic short datasets results in a significant decrease in accuracy. To mitigate this issue, we adopt LoRA-based fine-tuning, which allows the model to adapt to the short training data while keeping the accuracy loss relatively small.
>
> > **Q8:**  It seems that some important citations (esp. for the baseline methods) are missing, e.g. Laser paper.
>
> **A8:** We thank the reviewer for pointing this out. We have added the missing citations in the revised manuscript to ensure proper attribution and context.
>
> [1]Training Language Models to Reason Efficiently. 2025,11

---

> > ### Comment · Reviewer_tagM · 2025-11-25
> > **Thanks for the clarification**
> >
> > Dear authors,
> >
> > Thanks for giving a point-by-point rebuttals to my earlier questions/suggestions with concrete experiments. While there are certain questions remaining (see below), I feel that many of my concerns are addressed and I'm willing to **_increase my score_** to reflect the hard-works being done.
> >
> > Having said that, I still have two remaining concerns, and hope to hear what you think:
> >
> > 1.
> > > Distinguishing long from short incorrect trajectories does not add any useful information.
> >
> > Honestly, my sense is that these types of questions in "efficient LLM reasoning/addressing overthinking" literatures _can always be argued from opposite directions_. Your claim is that _"the model might intentionally produce short incorrect answers to obtain a reward comparable to that of long but correct trajectories, which would compromise accuracy"_, but another paper might say that incorrect reasonings also contain lots of redundant parts so penalizing long responses will seldom go wrong (actually many concurrent papers do apply length penalty to wrong answers as well).
> >
> > So my point is that maybe a (small-scale) ablation experiment can explain this better than any qualitative analysis.
> >
> > 2.
> >
> > Table 2 in the rebuttal above
> >
> > While I clearly see the detrimental effect of turning $w_l = 0$ (row 3) compared against $w_l = 1.0$ (row 1), I'm confused why the case with $w_l = 1.5, w_{acc} = 1.0$ (row 5) **has such a large trajectory length** -- i.e. the length penalty is clearly the largest, while the accuracy weight is similar to other cases. Is that a typo, or what could explain this observation (the accuracy also seems quite low)?

---

> > > ### Author Response · Authors · 2025-11-26
> > > **Response to Reviewer tagM Further Questions(1/2)**
> > >
> > > Dear Reviewer，
> > >
> > > Thank you very much for your thoughtful follow-up and for taking the time to re-evaluate our response. We address the new suggestions and questions you raised point by point below.
> > >
> > > >**Q9:** Maybe a (small-scale) ablation experiment can explain this (the reward function assigns a constant 0 length reward to all incorrect rollouts) better than any qualitative analysis.
> > >
> > > **R9:** We sincerely appreciate your valuable suggestions, which have helped improve our paper a lot. Following your suggestion, we conducted additional experiments with the DS-Distill-Qwen1.5B backbone, keeping all training parameters identical and **applying the length reward to all answers within each group (w/ incorrect)**. The results are shown in Table 3 and Table4. We will include these results and analysis in the updated version.
> > >
> > > We observe that when the length reward is applied to incorrect answers, the model achieves slightly higher accuracy compared to the Base model, but the decrease in token length is not substantial. In contrast, when the length reward is applied only to correct answers within the same group, the token length decreases more noticeably. This difference can be explained as follows:
> > >
> > > When the length reward is applied to incorrect answers, these answers, despite having an accuracy of 0, still receive some length reward. Since the model optimizes the total reward across all answers, the reward signals from incorrect answers interfere with the overall gradient direction and prevent the model from focusing solely on shortening correct answers (**short but incorrect answers** and **long but correct answers** receive similar total scores). To maximize the total reward, the model tends to generate longer outputs, including both correct and incorrect answers. This behavior can be seen as a form of *reward hacking*. In contrast, when the length reward is applied only to correct answers, the gradient signal is concentrated, allowing the model to shorten correct outputs as intended.
> > >
> > >
> > > **Table3. Comparison of Applying Length Reward to All Answers versus Correct Answers Only in In-Domain Tasks**
> > > |                      | GSM8K     |           | MATH500   |           | AIME24    |           | AMC       |           | Olympiad  |           | AVG       |           |
> > > |----------------------|-----------|-----------|-----------|-----------|-----------|-----------|-----------|-----------|-----------|-----------|-----------|-----------|
> > > |                      | ACC       | Length    | ACC       | Length    | ACC       | Length    | ACC       | Length    | ACC       | Length    | ACC       | Length    |
> > > | Base                 | 76.0      | 468       | 79.6      | 3617      | 28.7      | 11046     | 63.3      | 7644      | 47.0      | 7679      | 58.9      | 4765      |
> > > | w/o incorrect (ours) | 83.8      | 605       | 83.2      | 2072      | 32.0      | 6641      | 66.1      | 3872      | 51.6      | 4121      | **63.3**  | **2710**  |
> > > | w/ incorrect         | 78.0      | 871       | 82.4      | 3823      | 29.7      | 9711      | 64.2      | 7452      | 46.7      | 6997      | 60.2      | 4658      |
> > >
> > > **Table4. Comparison of Applying Length Reward to All Answers versus Correct Answers Only in Out-of-Domain Tasks**
> > > |                      | CSQA      |           | GPQA-Diamond |           | MMLU      |           | AVG       |           |
> > > |----------------------|-----------|-----------|--------------|-----------|-----------|-----------|-----------|-----------|
> > > |                      | ACC       | Length    | ACC          | Length    | ACC       | Length    | ACC       | Length    |
> > > | Base                 | 44.2      | 787       | 32.3         | 5619      | 43.8      | 1306      | 40.1      | 1370      |
> > > | w/o incorrect (ours) | 45.6      | 407       | 43.4         | 4182      | 44.6      | 715       | **44.5**  | **806**   |
> > > | w/ incorrect         | 44.9      | 502       | 36.1         | 5355      | 44.7      | 942       | 41.9      | 1040      |

---

> > > ### Author Response · Authors · 2025-11-26
> > > **Response to Reviewer tagM Further Questions(2/2)**
> > >
> > > > **Q10:** In Table 2, The length penalty is clearly the largest, while the accuracy weight is similar to other cases and large trajectory length. How to explain?
> > >
> > > **R10:** Thank you again for your constructive comments and for your engagement with our work. We sincerely appreciate it.
> > >
> > > As show in Table 2, we find that the trajectory becomes excessively long both when the length-reward weight is lower than the accuracy-reward weight (rows 3–4) and when it is higher (row 5). Although these two configurations result in similarly long trajectories, they arise from different forms of **reward hacking.** We clarify them below.
> > >
> > > **1. When $w_{\ell}$ < $w_{acc}$:**
> > >
> > > In this case, the model directly exploits the accuracy reward. Since longer reasoning traces typically increase the likelihood of receiving the accuracy score, the model quickly learns to produce longer outputs as a reliable strategy for obtaining high rewards. This is a standard and expected form of accuracy-driven reward hacking.
> > >
> > > **2. When $w_{\ell}$ > $w_{acc}$:**
> > >
> > > Even when the length reward is given a larger weight, the accuracy reward remains the dominant optimization signal. This is due to a scale mismatch between the two rewards:
> > >
> > > - the accuracy reward takes discrete values of 0 or 1, producing a large, sharp change in the return,
> > > - while the length reward provides a continuous value with a comparatively small range.
> > >
> > > As a result, the gradient induced by the accuracy reward is much stronger than that from the length reward, even when its explicit weight is smaller. To avoid the risk of receiving an accuracy reward of 0, the model again learns to generate longer outputs to obtain high accuracy reward. Hence, the model continues to exhibit reward-hacking behavior on the accuracy term.
> > >
> > > These observations demonstrate that an appropriate **balance between accuracy and length rewards is essential** for stable training and for preventing both forms of reward hacking. We will add more detailed explanation and discussion of these observations in Appendix C.4.

---

### Official Review · Reviewer_6BQF · 2025-10-31

**Soundness:** 2
**Presentation:** 3
**Contribution:** 3
**Rating:** 4
**Confidence:** 3

**Summary:**

Language Reasoning Models (LRMs) generally achieve higher accuracy as their reasoning depth increases; however, this improvement often comes at the cost of generating excessive tokens, which significantly slows down inference. Even when an LRM has already gathered sufficient information to reach the correct answer, it frequently continues producing unnecessary reasoning steps — a phenomenon referred to as overthinking.

This paper proposes a reinforcement learning framework called Just-Enough Thinking (JET), which enables LRMs to learn efficiently while suppressing overthinking. The core of JET lies in two main components: (1) a two-stage rollout construction, and (2) a reward design based on the principles of Correctness First, Conciseness Preference, and Per-Question Normalization.

In the two-stage rollout construction, JET uniformly truncates the model’s full reasoning trajectories at several intervals using a strategy called Progressive Early-Stopping (PES). Because PES utilizes trajectories generated directly by the model itself, it maintains the model’s natural consistency during training. Both full trajectories and truncated trajectories are then employed in reinforcement learning, allowing the model to learn when to stop reasoning without losing accuracy.

The reward function prioritizes correctness above all. Among responses that are correct, shorter reasoning paths are assigned higher rewards, thereby encouraging concise and efficient reasoning.

JET is applied to DeepSeek-Distill-Qwen-1.5B and 7B models and evaluated across five mathematical reasoning benchmarks — GSM8K, MATH500, AIME24, AMC, and Olympiad — as well as three out-of-domain datasets: CSQA, GPQA, and MMLU.

Across both in-domain and out-of-domain tasks, JET consistently improves accuracy while substantially reducing output token length, leading to faster inference compared with the base model.

The analysis further examines the effectiveness of the PES strategy and compares token length variations under different length reward designs, demonstrating the efficiency and robustness of the proposed method.

**Strengths:**

### 1. Overall Assessment
The proposed method, JET, is both simple and effective. Because it leverages trajectories directly generated by the model, JET preserves the model’s natural consistency during training. The approach is also computationally efficient, achieving faster training speeds without sacrificing accuracy.

### 2. Originality
The work presents a well-motivated application of Evidence Accumulation Theory to computational reasoning, highlighting meaningful parallels between human cognitive efficiency and model reasoning.
The design of the Progressive Early-Stopping (PES) mechanism and length-aware reinforcement learning is particularly well-executed. The reward function is carefully structured into three components — format reward, accuracy reward, and length reward — with the principles of Correctness First, Conciseness Preference, and Per-Question Normalization clearly reflected in the mathematical formulation.
Unlike previous approaches that rely on fixed token budgets or handcrafted truncation, JET autonomously learns when to stop reasoning in a self-consistent and context-sensitive manner.

### 3. Clarity
The paper is clearly written and well-organized, providing a strong motivation, detailed mathematical formulation, and a comprehensive algorithmic description (as presented in the Appendix). Figures such as Figure 1 (token length distribution) and Figure 2 (two-stage rollout diagram) effectively convey the core ideas and experimental insights.

**Weaknesses:**

### 1. Need for Further Discussion
Several aspects of the paper would benefit from deeper discussion. In the mathematical reasoning tasks, the Laser methods generally perform better on average and often outperform JET, achieving either higher accuracy or shorter token length. Moreover, Laser shows strong performance on out-of-domain datasets such as CSQA, GPQA, and MMLU. However, the paper provides almost no comparative discussion of these results, and notably, it does not cite the paper “Learn to Reason Efficiently with Adaptive Length-based Reward Shaping,” which originally proposed Laser-D/DE.
In the PES (Progressive Early-Stopping) strategy, trajectories are truncated arbitrarily at 25%, 50%, and 75% of their original length. The paper does not discuss the potential side effects of this truncation. While the authors state that JET preserves the model’s natural generation distribution, one could question whether cutting reasoning trajectories at arbitrary points might itself introduce unnatural discontinuities.
Finally, the paper lacks a discussion of its limitations — for instance, in scalability or potential failure cases.

### 2. Overstated Claims
Some of the claims in the paper appear somewhat overstated.
At line 400, the statement “the harder, the stronger” seems based on dataset-level observations and may be an overgeneralization. Since the study includes only eight datasets, this conclusion about performance trends appears insufficiently supported. An analysis at the sample-level difficulty would provide a stronger empirical basis for such a claim.
Additionally, while the paper states that JET’s generation process remains consistent with the model’s natural distribution, there is no experiment directly validating this claim. The intuition that JET should be “natural” is understandable. However, the paper does not include comparative experiments showing how other methods—for example, those using explicit length control or artificially shortened data—may lead to unnatural generation, nor how such unnaturalness affects performance, while JET maintains naturalness. Including such comparisons would strengthen the discussion.
In the 7B model, JET actually leads to accuracy drops on GSM8K and MATH500, which requires further analysis. Although the reward function is designed to favor correctness over length, these results suggest that this objective may not always hold in practice.

### 3. Experimental Scope
The study primarily evaluates JET on DeepSeek-Distill-Qwen models. To strengthen claims of generality, additional experiments on other architectures (e.g., Phi-4) would be desirable.

### 4. Minor Issues
Typographical errors: e.g., line 303, “we teste”; and in Equation (3), redundant parentheses “))”.

**Questions:**

Why did you not cite the Laser methods paper? The Laser methods generally show better average performance than the proposed JET approach on the target task (mathematical reasoning), and also perform strongly on out-of-domain benchmarks. In what aspects can JET be considered to have clear advantages over the Laser methods?

You argue that JET preserves the model’s natural reasoning distribution. Could you provide empirical validation for this claim? The Progressive Early-Stopping (PES) uses fixed truncation ratios (25%, 50%, 75%). Did you observe any cases where these arbitrary cutoffs caused the model to stop reasoning unnaturally — even if the truncated reasoning still led to correct answers?

Experiments are limited to the DeepSeek-Distill-Qwen models. Have you tested or considered applying JET to other models (e.g., Phi-4)?

How does the training time of JET compare to that of the Laser methods?

Finally, could you elaborate on potential scenarios where JET underperforms or fails?

---

> ### Author Response · Authors · 2025-11-24
> **Response to Reviewer 6BQF (1/2)**
>
> Thank you for your constructive comments and suggestions. We have revised our paper according to your comments. We respond to your questions below and would appreciate it if you could let us know if our response addresses your concerns.
>
> > **Q1:** The Laser methods generally show better average performance than the proposed JET approach on the target task (mathematical reasoning), and also perform strongly on out-of-domain benchmarks. In what aspects can JET be considered to have clear advantages over the Laser methods?
>
> **A1:**  We summarize JET’s advantages over Laser from three perspectives: (1) stable in-domain performance across tasks of varying difficulty, (2) stronger generalization on out-of-domain benchmarks, and (3) lower training cost and simpler implementation.
>
> 1. **In-domain performance:** On the 1.5B backbone, JET outperforms Laser in both accuracy and output length. While JET shows slightly lower accuracy than Laser-DE on the 7B backbone, it exhibits more stable performance across tasks of varying difficulty. For example, on simpler tasks such as GSM8K, Laser-DE significantly increases token usage (+105.8%), suggesting that its dynamic difficulty-aware mechanism does not adapt flexibly. In contrast, JET consistently balances accuracy and output length across all task difficulties.
>
> 2. **Out-of-domain performance:** As reported in Table 2, **JET outperforms Laser on both backbones**:
>
> - 1.5B backbone: Laser-DE accuracy 42.9, avg. tokens 1223; JET accuracy 44.5, avg. tokens 806.
>
> - 7B backbone: Laser-DE accuracy 59.7, avg. tokens 1090; JET accuracy 60.9, avg. tokens 1013.
>
> Although performances are close on some datasets, JET consistently outperforms Laser on most tasks. Pareto plots (Figure 8) comparing accuracy and average output token length further illustrate JET’s advantage, and additional analyses have been included in the updated manuscript Section 4.2.
>
> 3. **Training cost and operational simplicity:** Beyond performance, JET is simpler and more efficient to implement. It constructs the short training samples required for RL by truncating full reasoning trajectories during the rollout phase. In contrast, Laser dynamically adjusts the target length based on problem difficulty, increasing rollout time. For example, with the 1.5B backbone on the same data, average rollout times are: DAPO 800s, JET 220s, and Laser 1100s.
>
> In summary, although the results on the 7B model show similar results for JET and Laser, JET demonstrates clear advantages in stability, generalization, and efficiency. We have revised the manuscript to provide a more detailed analysis of these points.
>
> > **Q2:** Additionally, while the paper states that JET’s generation process remains consistent with the model’s natural distribution, there is no experiment directly validating this claim. The intuition that JET should be “natural” is understandable. However, the paper does not include comparative experiments showing how other methods—for example, those using explicit length control or artificially shortened data—may lead to unnatural generation, nor how such unnaturalness affects performance, while JET maintains naturalness. Including such comparisons would strengthen the discussion.
>
> **A2:**  Thank you for recognizing that our truncation strategy is intuitively consistent with preserving the model’s natural generation distribution, and for suggesting that we provide additional justification. Your feedback is very helpful for improving the clarity of our paper. We address this point from both theoretical and empirical perspectives.
>
> 1. Theoretical/Design Perspective: In JET training, rollouts follow a two-stage construction: full reasoning trajectories are first preserved, and truncation is then applied only to model-generated trajectories, with a minimal number of stop-thinking tokens appended to guide the final answer. Most of each trajectory remains drawn from the model’s natural distribution, ensuring that the truncated samples stay close to the original policy. Consequently, their log-prob ratios remain stable, preventing clipping from discarding them and allowing the advantage terms to contribute effective gradients. A detailed analysis of the gradient impact of truncated trajectories under the GRPO framework is provided in Section 3.3.
>
> 2. Empirical Perspective: We provide additional evidence by examining the KL-loss dynamics during training. Using the DS-Qwen-1.5B backbone, we compare the KL trajectories of JET and the DAPO baseline. The KL curves show that JET does not deviate from the model’s native distribution and in fact maintains a lower KL compared to the baseline DAPO setup, indicating that our truncation does not introduce distributional distortion. The full KL-loss results and discussion have been added to Section 4.5.

---

> ### Author Response · Authors · 2025-11-24
> **Response to Reviewer 6BQF (2/2)**
>
> >**Q3:** Did you observe any cases if the truncated reasoning still led to correct answers?
>
> **A3:** Thank you for the question. In our pilot experiments, truncating the reasoning at different positions can still yield correct answers while significantly reducing token usage. More importantly, during RL training, the full reasoning trajectory is always included in the rollout (Figure 1b and 1c). This ensures that even if all truncated versions are incorrect, the full trajectory receives the highest reward and dominates the learning signal. Consequently, truncation does not prevent the model from learning correct reasoning behavior.
>
> > **Q4:** Experiments are limited to the DeepSeek-Distill-Qwen models. Have you tested or considered applying JET to other models (e.g., Phi-4)?
>
> **A4:** We thank the reviewer for the suggestion. The two backbones we initially selected are commonly used in prior work on efficient reasoning. To further validate JET’s performance, we added DeepSeek-R1-Distill-Llama-8B as a backbone, with results shown in Table 1. Since the referenced baselines did not use this backbone, we report only a subset of them.
>
> We find that JET demonstrates strong generalization across backbones, maintaining a stable balance between accuracy and output length. Quantitatively, compared to the base model, JET improves accuracy by 2 points while reducing average token usage by 31%. On out-of-domain tasks, it reduces token usage by 29% while gaining 0.8 points in accuracy. Detailed experimental results and further analyses are provided in the appendix C.2.
>
> **Table 1. The comparison performance on DeepSeek-Distill-Llama-8B.**
> | Methods | GSM8K |  | MATH500 |  | AIME24 |  | AMC |  | Olympiad |  | AVG |  |
> | :--- | :--- | :--- | :--- | :--- | :--- | :--- | :--- | :--- | :--- | :--- | :--- | :--- |
> |  | ACC | Length | ACC | Length | ACC | Length | ACC | Length | ACC | Length | ACC | Length |
> | Base | 88.6 | 1756 | 88.0 | 4299 | 41.0 | 12223 | 76.9 | 7521 | 52.4 | 8193 | 69.4 | 5493 |
> | SFT | 69.5 | 410 | 77.6 | 2833 | 34.7 | 9683 | 72.8 | 6392 | 47.3 | 6249 | 60.4 | 3970 |
> | DPO | 71.0 | 415 | 78.6 | 3215 | 36.3 | 11314 | 76.4 | 6450 | 49.5 | 6868 | 62.4 | 4288 |
> | DAPO | 81.7 | 1027 | 86.6 | 3258 | 40.8 | 9701 | 77.4 | 5463 | 52.4 | 6463 | 67.8 | 4081 |
> | JET | 91.1 | 1002 | 89.0 | 2922 | 41.0 | 9231 | 76.8 | 5261 | 59.1 | 5822 | **71.4** | **3821** |
>
> > **Q5:** At line 400, the statement “the harder, the stronger” seems based on dataset-level observations and may be an overgeneralization.
>
> **A5:** Thank you for the suggestion. We have revised the original sentence as follows:
> _The better performance of JET on difficult tasks indicates that JET enhances the model’s ability to capture deep reasoning structures, rather than relying on superficial pattern matching, and maintains high performance even in complex domains._
>
> >**Q6:** In the 7B model, JET shows slight accuracy drops on GSM8K and MATH500. Could you clarify why this occurs, given that the reward function prioritizes correctness?
>
> **A6:** Thank you for the question. On the 7B model, slight accuracy drops are observed on GSM8K and MATH500 (-0.9 and -0.8, respectively), which can be attributed to the relatively simple nature of these tasks compared to other complex math benchmarks. JET’s gains are more pronounced on more difficult problems, so minor fluctuations on simpler tasks are expected and do not indicate a flaw in the reward design.
>
> >**Q7:** How does the training time of JET compare to that of the Laser methods?
>
> **A7:**  Regarding training efficiency, JET is significantly faster than Laser methods. For example, using the 1.5B backbone on the same training data, the average rollout times are: DAPO 800s, JET 220s, and Laser 1100s. This demonstrates that JET not only achieves competitive performance but also reduces training time and computational cost compared to Laser.
>
> >**Q8:** Could you elaborate on potential scenarios where JET underperforms or fails?
>
> **A8:** Thank you for the question. JET may be less effective in scenarios where the reasoning process is inherently dense, meaning that nearly every step contributes essential information to the final answer. In such cases, truncation provides limited opportunity to shorten the trajectory, so JET may not yield substantial token savings.
>
> However, this limitation is not unique to JET. When a task truly requires a fully detailed chain-of-thought, any method that attempts to compress or restructure the reasoning, such as explicit length control or aggressive truncation, would face the same difficulty.

---

### Official Review · Reviewer_Zu3p · 2025-10-31

**Soundness:** 2
**Presentation:** 2
**Contribution:** 2
**Rating:** 4
**Confidence:** 4

**Summary:**

The paper proposes a method called Just-Enough Thinking (JET) that trains LRMs to proactively terminate unnecessary reasoning. Specifically, the paper first performs pilot experiments to reveal that LRMs have accumulated sufficient information early in their reasoning. Based on this phenomenon, JET adopts a Progress Early-Stopping (PES) strategy to sample reasoning trajectories with varying length and introduces a linearly normalized length reward to encourage more efficient reasoning. Experimental results show that JFT significantly improves reasoning efficiency without sacrificing accuracy on 5 math benchmarks and some other reasoning benchmarks.

**Strengths:**

1.The paper is mostly well written and presents the topic clearly.

2.The pilot studies are interesting and directly supports the design of the JET method.

3.The proposed method (JET) is described in sufficient detail and appears technically sound.

**Weaknesses:**

Major Concerns

1.Potential distributional inconsistency. The paper introduces a truncation strategy that also involves inserting an additional prompt. There is insufficient evidence to confirm that this modification does not introduce a harmful distributional inconsistency problem.

2.Sub-optimal performance on math tasks. The results on mathematical reasoning tasks do not clearly demonstrate the superiority of the proposed method. As shown in Table 1, the JFT-DeepSeek-Distill-Qwen-7B model underperforms the Laser-DE model in both accuracy (-0.5%) and efficiency (using more tokens).

3.Insufficient experiments. The empirical evaluation is constrained to a specific set of models (DeepSeek-Distill-Qwen-1.5B and DeepSeek-Distill-Qwen-7B). The experiments should be extended to include more recent model families (e.g., the Qwen3 series) and potentially more model sizes.

**Questions:**

My main questions for the authors directly relate to the major concerns identified above. Addressing these points during the rebuttal period would be crucial for re-evaluating my rating.

---

> ### Author Response · Authors · 2025-11-24
> **Response to Reviewer Zu3p**
>
> We sincerely thank the reviewer for the constructive feedback and the detailed assessment of our work. We have revised our paper based on your feedback. We detail our response below and please kindly let us know if our response addresses your concerns.
>
> > **Q1**: Potential distributional inconsistency. The paper introduces a truncation strategy that also involves inserting an additional prompt. There is insufficient evidence to confirm that this modification does not introduce a harmful distributional inconsistency problem.
>
> **A1:** Thank you for the comment. We would like to clarify that JET does not introduce harmful distributional inconsistency, from both theoretical and empirical perspectives.
>
> 1. Theoretical / Design Perspective: In JET training, rollouts follow a two-stage construction: full reasoning trajectories are first preserved, and truncation is then applied only to model-generated trajectories, with a minimal number of stop-thinking tokens appended to guide the final answer. Most of each trajectory remains drawn from the model’s natural distribution, ensuring that the truncated samples stay close to the original policy. Consequently, their log-prob ratios remain stable, preventing clipping from discarding them and allowing the advantage terms to contribute effective gradients. A detailed analysis of the gradient impact of truncated trajectories under the GRPO framework is provided in Section 3.3.
>
> 3. Empirical Perspective: We also include empirical evidence from KL-loss dynamics during training. The KL curves, as shown in Figure 3b, show that JET does not deviate from the model’s native distribution and in fact maintains a lower KL compared to the baseline DAPO setup, indicating that our truncation does not introduce distributional mismatch. The full KL-loss results and discussion have been added to the Section 4.1.
>
> Overall, both the design and empirical evidence indicate that JET preserves the model’s natural generation distribution and maintains stable gradient updates, addressing potential concerns regarding distributional inconsistency.
>
> > **Q2:** Sub-optimal performance on math tasks. The results of mathematical reasoning tasks do not clearly demonstrate the superiority of the proposed method. As shown in Table 1, the JFT-DeepSeek-Distill-Qwen-7B model underperforms the Laser-DE model in both accuracy (-0.5%) and efficiency (using more tokens).
>
> **A2:** While JET shows slightly lower accuracy on the 7B backbone compared to Laser-DE, it demonstrates more stable performance across tasks of varying difficulty. For example, on simpler tasks like GSM8K, Laser significantly increases token usage (+105.8%), indicating that its dynamic difficulty-aware mechanism does not adapt flexibly across tasks of different difficulty. In contrast, JET consistently balances accuracy and output length across all task difficulties. Besides, comparing accuracy and average output token length for JET and other baselines, showing that **JET achieves advantages on most tasks.** Additional analysis of these performance comparisons has been included in the updated version of the paper.
>
> > **Q3:** Insufficient experiments. The empirical evaluation is constrained to a specific set of models, including DS-Distill-Qwen-1.5B and 7B. The experiments should be extended to include more recent model families and potentially more model sizes.
>
> **A3:** We thank the reviewer for the suggestion. The two backbones we initially selected are commonly used in prior work on efficient reasoning. To further validate JET’s performance, we added DeepSeek-R1-Distill-Llama-8B as a backbone, with results shown in Table 1. Since the referenced baselines did not use this backbone, we report only a subset of them.
>
> We find that JET demonstrates strong generalization across backbones, maintaining a stable balance between accuracy and output length. Quantitatively, compared to the base model, JET improves accuracy by 2 points while reducing average token usage by 31%. On out-of-domain tasks, it reduces token usage by 29% while gaining 0.8 points in accuracy. Detailed experimental results and further analyses are provided in the appendix C.2.
>
> **Table 1. The comparison performance on DeepSeek-Distill-Llama-8B.**
> | Methods | GSM8K |  | MATH500 |  | AIME24 |  | AMC |  | Olympiad |  | AVG |  |
> | :--- | :--- | :--- | :--- | :--- | :--- | :--- | :--- | :--- | :--- | :--- | :--- | :--- |
> |  | ACC | Length | ACC | Length | ACC | Length | ACC | Length | ACC | Length | ACC | Length |
> | Base | 88.6 | 1756 | 88.0 | 4299 | 41.0 | 12223 | 76.9 | 7521 | 52.4 | 8193 | 69.4 | 5493 |
> | SFT | 69.5 | 410 | 77.6 | 2833 | 34.7 | 9683 | 72.8 | 6392 | 47.3 | 6249 | 60.4 | 3970 |
> | DPO | 71.0 | 415 | 78.6 | 3215 | 36.3 | 11314 | 76.4 | 6450 | 49.5 | 6868 | 62.4 | 4288 |
> | DAPO | 81.7 | 1027 | 86.6 | 3258 | 40.8 | 9701 | 77.4 | 5463 | 52.4 | 6463 | 67.8 | 4081 |
> | JET | 91.1 | 1002 | 89.0 | 2922 | 41.0 | 9231 | 76.8 | 5261 | 59.1 | 5822 | **71.4** | **3821** |

---

> > ### Comment · Reviewer_Zu3p · 2025-11-27
> >
> > Thank you for the clarifications. While Q1 and Q3 have been well addressed, concerns regarding Q2 remain. Could you further clarify the reasons for the inferior performance and discuss how this impacts the usefulness in real-world practice?

---

> > > ### Author Response · Authors · 2025-11-27
> > > **Further Clarification on Q2 from Reviewer Zu3p**
> > >
> > > Dear Reviewer，
> > >
> > > We sincerely appreciate your feedback. We have provided a more detailed clarification regarding Q2.
> > >
> > > > **Q2:** Sub-optimal performance on math tasks. The results of mathematical reasoning tasks do not clearly demonstrate the superiority of the proposed method. As shown in Table 1, the JFT-DeepSeek-Distill-Qwen-7B model underperforms the Laser-DE model in both accuracy (-0.5%) and efficiency (using more tokens)
> > >
> > > **A2:** We summarize JET’s advantages over Laser from three perspectives: (1) stable in-domain performance across tasks of varying difficulty, (2) stronger generalization on out-of-domain benchmarks, and (3) lower training cost and simpler implementation.
> > >
> > > 1. **In-domain performance:** On the 1.5B backbone, JET outperforms Laser in both accuracy and output length. While JET (acc:75.2) shows slightly lower accuracy than Laser-DE (acc:76.9) on the 7B backbone, it exhibits more stable performance across tasks of varying difficulty. For example, on simpler tasks such as GSM8K, **Laser-DE** significantly increases token usage (**+105.8%**), suggesting that its dynamic difficulty-aware mechanism does not adapt flexibly. In contrast, **JET** reduces token usage (**-30.9%**), demonstrating a consistent balance between accuracy and efficiency across all task complexities.
> > >
> > > 2. **Out-of-domain performance:** As reported in Table 2, **JET outperforms Laser on both backbones**:
> > >
> > > - 1.5B backbone: Laser-DE accuracy 42.9, avg. tokens 1223; JET accuracy 44.5, avg. tokens 806.
> > >
> > > - 7B backbone: Laser-DE accuracy 59.7, avg. tokens 1090; JET accuracy 60.9, avg. tokens 1013.
> > >
> > > Although performances are close on some datasets, JET consistently outperforms Laser on most tasks. Pareto plots (Figure 8) comparing accuracy and average output token length further illustrate JET’s advantage, and additional analyses have been included in the updated manuscript Section 4.2.
> > >
> > > 3. **Training cost and operational simplicity:** Beyond performance, JET is simpler and more efficient to implement. It constructs the short training samples required for RL by truncating full reasoning trajectories during the rollout phase. In contrast, Laser dynamically adjusts the target length based on problem difficulty, increasing rollout time. For example, with the 1.5B backbone on the same data, average rollout times are: **DAPO 800s, JET 220s**, and **Laser 1100s.**
> > >
> > > In summary, although the results on the 7B model show similar results for JET and Laser, JET demonstrates clear advantages in stability, generalization, and efficiency. We have revised the manuscript to provide a more detailed analysis of these points.
> > >
> > > Thanks again for your feedback! We hope the clarification above satisfactorily addresses your concerns. If you have any further questions, we would be happy to clarify or provide additional details.

---

> > > > ### Comment · Reviewer_Zu3p · 2025-11-28
> > > >
> > > > Thanks for your detailed response and my concern has been addressed. I plan to increase my score to reflect this; however, I am currently unable to edit my review due to OpenReview system restrictions. I am leaving a note here and will update the rating once this issue is resolved.

---

### Official Review · Reviewer_A4Dw · 2025-11-01

**Soundness:** 2
**Presentation:** 3
**Contribution:** 2
**Rating:** 4
**Confidence:** 4

**Summary:**

This paper proposes JET, a reinforcement learning method based on DAPO to reduce unnecessary reasoning steps in Large Reasoning Models. The approach uses Progressive Early-Stopping to create truncated reasoning trajectories during training and employs a length-aware reward to encourage concise outputs. Experiments on DeepSeek-Distill-Qwen models (1.5B and 7B) show token reductions while maintaining accuracy on math benchmarks.

**Strengths:**

1. The paper demonstrates that LRMs accumulate sufficient information early in reasoning chains, providing good empirical justification for the approach.

2. JET generates shortened trajectories from the model's sampled outputs, reducing distribution mismatch issues that arise from using external short answers.

3. The paper includes  evaluations across multiple benchmarks and model sizes.

**Weaknesses:**

1. In Section 1, line 87, I think that assigning length penalties to longer correct trajectories is problematic because the model cannot guarantee that longer sequences always contain redundant information. Although the authors use a "Correctness first" rule where shorter responses receive better reward signals, penalizing longer correct answers does not make much sense.

2. The values of non-negative coefficients w_f, w_acc, and w_ℓ are shown in the appendix as w_acc = 0.9, w_f = 0.1, and w_ℓ = 1. More diverse strategies should be explored since these coefficients significantly influence performance.

3. Since JET is implemented based on DAPO, the accuracy of DAPO should theoretically represent the upper bound for JET. However, I am confused why GSM8K improves by 3.8 points when length decreases from 826 to 605, AIME24 improves by 6 points when length decreases from 8000 to 6000, and Olympiad improves by 5 points when length decreases from 5000 to 4000.

4. When the model scales from 1.5B to 7B, most baseline models decrease or slightly increase sequence length on AIME24, AMC, and Olympiad. However, JET increases sequence length from 6641 to 7981 on AIME24, 3872 to 4301 on AMC, and 4121 to 5083 on Olympiad. What explains this counterintuitive behavior?

5. In Table 2 regarding language reasoning tasks, JET's sequence length is 1013, which is comparable to other baseline models, indicating that JET does not achieve expected length reduction on these tasks.

6. In Table 2, why do baseline models decrease length when scaling from 1.5B to 7B, while JET increases from 800 to 1000? More analysis should be provided here.

7. The paper presents performance on many benchmarks but lacks analysis of model training dynamics such as KL, policy entropy, gradient norm, etc.

8. In Section 3.2, there is ambiguity: does "ℓ_min and ℓ_max are the shortest and longest correct responses" refer to the responses themselves or their lengths?

9. In Formula 6, what would happen if the "+ δ" term were removed?

**Questions:**

See the Weaknesses described above.

---

> ### Author Response · Authors · 2025-11-24
> **Response to Reviewer A4Dw(1/2)**
>
> Thank you for your valuable feedback to help us improve our paper. We have revised our paper based on your feedback. We detail our response below and please kindly let us know if our response addresses your concerns.
>
> > **Q1:** In Section 1, line 87, I think that assigning length penalties to longer correct trajectories is problematic because the model cannot guarantee that longer sequences always contain redundant information.
>
> **A1:** We thank the reviewer for the comment and would like to clarify the design of our length reward. **The length reward is applied only among correct answers for the same question.** The shortest correct trajectory is treated as minimal sufficient reasoning, and other correct trajectories receive a smaller reward. For example, if question q yields three answers A, B, and C, where both A and B are correct and B is the shortest, A receives a smaller length reward than B. This design preserves correctness as the primary objective while encouraging conciseness. We have revised the sentence in line 87 to further clarify this:
>
> _"we introduce a quality-controlled length reward to better guide the model toward efficiency. For each reasoning group, if multiple trajectories reach the correct answer, we use the shortest correct trajectory as a reference and assign a length reward to all correct trajectories, where the reward decreases gradually with trajectory length."_
>
> > **Q2:** The appendix lists the reward coefficients as w_acc = 0.9, w_f = 0.1, and w_ℓ = 1. Because these coefficients strongly affect model behavior, more diverse weight settings should be explored to better understand their impact.
>
> **A2:** The format reward plays only a minimal and expected role and is not the primary focus of our analysis. We do not conduct a separate analysis on it. In our setup, the accuracy reward weight (w_acc = 0.9) and the format reward weight (w_f = 0.1) follow standard practice in RLVR [1,2].
>
> Following your suggestion, we conduct additional experiments to study how the accuracy and length reward coefficients (w_acc and w_ℓ) influence JET. Using DS-Distill-Qwen-1.5B as the backbone and keeping all other RL settings fixed, we report the results in the table 1 below. Our findings are as follows：
>
> First, our length reward effectively encourages concise reasoning. Although PES produces short trajectories during rollout stage, it does not modify the model’s preference; without the length reward, the model shifts toward much longer generations.
>
> Second, when $w_{acc}$ heavily dominates $w_ℓ$, the model directly exploits the accuracy reward and output longer answer. Since longer reasoning traces typically increase the likelihood of receiving the accuracy score, the model quickly learns to produce longer outputs as a reliable strategy for obtaining high rewards. This is a standard and expected form of accuracy-driven reward hacking.
>
> Third,  even when the length reward is given a larger weight, the accuracy reward remains the dominant optimization signal. This is due to a scale mismatch between the two rewards:
>
> * the accuracy reward takes discrete values of 0 or 1, producing a large, sharp change in the return,
> * while the length reward provides a continuous value with a comparatively small range.
>
> As a result, the gradient induced by the accuracy reward is much stronger than that from the length reward, even when its explicit weight is smaller. To avoid the risk of receiving an accuracy reward of 0, the model again learns to generate longer outputs to obtain high accuracy reward. Hence, the model continues to exhibit reward-hacking behavior on the accuracy term.
>
> These observations demonstrate that an appropriate balance between accuracy and length rewards is essential for stable training and for preventing both forms of reward hacking. The detailed discussion and additional analysis of this part have been included in **Appendix C.4**.
>
> **Table1: Effect of accuracy $w_{acc}$ and length $w_{\ell}$ reward coefficients on model performance.**
> | Methods | GSM8K |  | MATH500 |  | AIME24 |  | AMC |  | Olympiad |  | AVG |  |
> | :--- | :--- | :--- | :--- | :--- | :--- | :--- | :--- | :--- | :--- | :--- | :--- | :--- |
> |  | ACC | Length | ACC | Length | ACC | Length | ACC | Length | ACC | Length | ACC | Length |
> | Base | 76.0 | 468 | 79.6 | 3617 | 28.7 | 11046 | 63.3 | 7644 | 47.0 | 7679 | 58.9 | 4765 |
> | $w_{\ell} = 1.0, w_{acc} = 0.9$ | 83.8 | 605 | 83.2 | 2072 | 32.0 | 6641 | 66.1 | 3872 | 51.6 | 4121 | **63.3** | **2710** |
> | $w_{\ell} = 0.9, w_{acc} = 1.0$ | 82.6 | 726 | 82.2 | 2459 | 28.7 | 7731 | 66.0 | 4783 | 49.3 | 4852 | 61.8 | 3243 |
> | $w_{\ell} = 0.0, w_{acc} = 0.9$ | 84.2 | 886 | 84.0 | 2997 | 29.3 | 9822 | 65.9 | 5671 | 48.7 | 6044 | 62.4 | 3974 |
> | $w_{\ell} = 1.0, w_{acc} = 1.5$ | 81.4 | 985 | 85.4 | 3591 | 30.7 | 10332 | 66.8 | 6727 | 47.7 | 6963 | 62.4 | 4547 |
> | $w_{\ell} = 1.5, w_{acc} = 1.0$ | 79.8 | 981 | 83.4 | 3553 | 25.7 | 10532 | 64.1 | 6406 | 48.6 | 6959 | 60.3 | 4482 |

---

> ### Author Response · Authors · 2025-11-24
> **Response to Reviewer A4Dw(2/2)**
>
> >**Q3:** Since JET builds on DAPO, DAPO’s accuracy might be expected to represent an upper bound. Why then does JET improve on GSM8K, AIME24, and Olympiad when reasoning length decreases?
>
> **A3:** **The performance of DAPO does not represent a upper bound for JET.** While DAPO/GRPO training often produces longer reasoning trajectories[3][4], longer responses do not always lead to higher accuracy. Excessively long reasoning can introduce redundant or erroneous steps, whereas more concise trajectories focus on essential reasoning steps, which can improve performance on certain problems. Besides, in our experiments, other baselines trained with DAPO/GRPO also surpass the original DAPO on some tasks.  For instance, with the 1.5B model, AdaThink (accuracy: 49.8) surpasses DAPO (accuracy: 46.6) on the Olympiad dataset. These observations indicate that JET achieving higher accuracy than DAPO on some tasks is expected behavior.
>
> > **Q4:** When scaling from 1.5B to 7B, most baselines shorten or slightly shorten their responses on AIME24, AMC, and Olympiad, but JET produces longer outputs. Similarly, in Table 2, baselines become shorter when scaling, while JET increases length. More analysis is needed to explain this behavior.
>
> **A4:** Thank you for pointing out this phenomenon. The increase in average length after scaling is not anomalous. We analyze JET-1.5B and JET-7B by separately measuring the average token lengths of correct and incorrect outputs. The results are shown in Table 2. Our findings are:
>
> (1) For correct answers, there is no consistent trend when scaling from 1.5B to 7B. The sequences become longer on some tasks and shorter on others. Baseline models exhibit the same non-monotonic pattern, indicating that the change is not a systematic effect of model size.
>
> (2) For incorrect answers, JET-7B consistently produces longer responses than JET-1.5B, suggesting that larger models tend to generate longer reasoning chains when they are uncertain.
>
> **Table2. Average output token length of JET-7B and JET-1.5B for correct and incorrect answers.**
> | Dataset | Correct Answer |  | Incorrect Answer |  |
> | :--- | :--- | :--- | :--- | :--- |
> |  | 7B | 1.5B | 7B | 1.5B |
> | GSM8K | 518 | 534 | 1092 | 1022 |
> | MATH500 | 1490 | 1393 | 6700 | 4646 |
> | AIME24 | 4765 | 5089 | 10693 | 8147 |
> | AMC | 3365 | 2627 | 9518 | 6214 |
> | Olympiad | 2868 | 2142 | 7876 | 6152 |
> | CSQA | 344 | 371 | 452 | 430 |
> | GPQA | 3825 | 3957 | 6109 | 4198 |
> | MMLU | 758 | 696 | 1016 | 731 |
> | AVG | 1460 | 1286 | 3580 | 2638 |
>
> > **Q5：** In Table 2, JET reaches a sequence length of about 1000 on language reasoning tasks, similar to other baselines. This suggests that JET does not reduce length as expected on these tasks.
>
> **A5:**  Thank you for the comment. First, although JET’s output lengths on these language reasoning tasks are comparable to those of the baselines, it achieves the highest average accuracy, demonstrating its effectiveness. Second, the similar token lengths across all baselines suggest that these tasks offer limited scope for further compression; maintaining outputs of this length is necessary to preserve high accuracy.
>
> > **Q6:** The paper presents performance on many benchmarks but lacks analysis of model training dynamics such as KL, policy entropy, gradient norm, etc.
>
> **A6:**  Thank you for the suggestion. In our experiments, we track several dynamic metrics to analyze model training behavior:
>
> （1）KL loss, which demonstrates that JET preserves the model’s natural probability distribution during RL training.
>
> （2）Rollout time and training time, which illustrate how our truncation strategy accelerates RL training.
>
> （3）Training dynamics of each benchmark, including accuracy and average output length throughout training.
>
> While metrics (2) and (3) were already reported in the original paper in the Figure 4,6,7, the updated version includes analysis of metric (1) in Figure 3b.
>
> > **Q7:** In Section 3.2, there is ambiguity: does "ℓ_min and ℓ_max are the shortest and longest correct responses" refer to the responses themselves or their lengths?
>
> **A7:** Here, ℓ_min and ℓ_max denote the lengths of the shortest and longest trajectories among the correct responses within a reasoning group. We have updated the main text to make this interpretation explicit and avoid ambiguity.
>
> > **Q8:** In Formula 6, what would happen if the "+ δ" term were removed?
> >
> **A8:**  If the “+ δ” term were removed, correct responses that are slightly longer would receive rewards that approach zero as their lengths get close to ℓ_max. The δ term ensures a non-zero lower bound, the correct responses receive at least δ length reward.
>
> [1] ReTool: Reinforcement Learning for Strategic Tool Use in LLMs
>
> [2]Learning to Reason under Off-Policy Guidance.
>
> [3] DAPO: An Open-Source LLM Reinforcement Learning System at Scale. 2025, NIPS.
>
> [4] DeepSeekMath: Pushing the Limits of Mathematical Reasoning in Open Language Models. 2024, Arxiv

---

### Comment · Area_Chair_dL22 · 2025-11-26

Dear Reviewers,

Would you please check authors' rebuttal and see if they have addressed your comments?

Best

AC

---

### Author Response · Authors · 2025-12-02
**Final Response [1/2]**

We sincerely thank the new Area Chair for evaluating our submission under the unexpected circumstances caused by the recent OpenReview incident. We also appreciate the original Area Chair and the reviewers for their careful reading and constructive feedback. We understand the additional workload created by the situation and are grateful for the opportunity to provide this final response.

**A.** First, we thank all reviewers for their positive assessments. The main strengths acknowledged across reviews are:

- The pilot studies provide an intuitive and compelling justification for JET,  providing strong empirical support for the approach (Reviewer A4Dw, Zu3p, and tagM).
- JET is both simple and effective. By generating shortened trajectories directly from the model’s outputs, JET reduces distribution mismatch issues associated with using external short answers, while preserving the model’s natural consistency and achieving computational efficiency (Reviewer A4Dw, 6BQF).
- The experimental evaluation is comprehensive, covering multiple benchmarks, diverse in-domain and out-of-domain tasks, various model sizes, and comparisons with alternative efficient reasoning baselines (Reviewer A4Dw, tagM).
- The paper is clearly written, well-organized, and easy to follow, with detailed mathematical formulations, algorithmic descriptions, illustrative figures that effectively convey core ideas and experimental insights, and  the proposed method is described in sufficient technical detail and appears technically sound (Reviewer Zu3p, 6BQF, tagM).
- The originality of our method is recognized, particularly in the design of Progressive Early-Stopping (PES), length-aware reinforcement learning, and a carefully structured reward function that balances correctness, conciseness, and per-question normalization, reflecting meaningful parallels with human cognitive efficiency (Reviewer 6BQF).

---

### Author Response · Authors · 2025-12-02
**Final Response [2/2]**

**B.** We acknowledge the reviewers’ concerns, which we have addressed in detail in the reviewer-specific Official Comments. Below, we provide a concise summary of each reviewer’s main concerns, our responses, the corresponding revisions in the paper, and related communications with the reviewers. Additionally, we have uploaded an updated version of the paper, with all modifications highlighted in dark blue for easy reference.


| Reviewers | Main Concern | Our Response / Revisions | Discuss Details |
| :--- | :--- | :--- | :--- |
| **A4Dw** | 1. The paper does not provide an analysis of the weighting coefficients for the three reward scores. (W2) | 1. We first clarify that the *format* reward has only a minor effect. We then conduct additional experiments to examine how the weighting coefficients of the *length* reward and *accuracy* reward influence model performance. The corresponding results and analyses are provided in Appendix C.4. | No feedback from the reviewers has been received so far. |
|  | 2. Several questions are specific or relatively trivial. (Excluding W2) | 2. We address each of the specific points raised by the reviewer in detail. |  |
|**Zu3p**| 1. Although the intuition that truncation aligns with the policy model’s probability distribution is reasonable, the paper lacks experiments to validate this claim. (W1) | 1. We clarify that the truncated answers produced by our method are well aligned with the model’s probability distribution, supported by both **theoretical analysis** and **empirical evidence**. Specifically, we have added a theoretical analysis from the perspective of gradient updates in Section 3.1, and an empirical analysis based on KL-divergence dynamics in Section 4.1. | After two rounds of discussion, the reviewer confirmed that our responses have addressed his/her concerns and expressed willingness to raise the score. However, the system does not currently allow score updates, so the reviewer left a separate note in the Comments. |
|  | 2. The performance advantage of JET over Laser is not sufficiently highlighted. (W2) | 2. We further clarify the advantages of JET by demonstrating its **stable in-domain performance** across tasks of varying difficulty, its **stronger generalization on out-of-domain** benchmarks, and its **lower training cost coupled with a simpler implementation**. | |
|  | 3. The evaluation is limited beyond Deepseek-Distilled-Qwen. (W3) | 3. We have added experiments evaluating JET on models beyond the Qwen series to verify its generality. The corresponding results and analyses are provided in Appendix C.2. |  |
|**6BQF** | Same as reviewer Zu3p. | Same as above | No feedback from the reviewers has been received so far. |
| **tagM** | 1. The evaluation is limited beyond Deepseek-Distilled-Qwen. (W1) | 1. We have added experiments evaluating JET on models beyond the Qwen series to verify its generality. The corresponding results and analyses are provided in Appendix C.2. | The reviewer indicated that our responses have satisfactorily addressed his/her concerns and updated the score to 6. |
|  | 2. "Overthinking" also includes case (ii): overly long or convoluted reasoning that ultimately leads to incorrect answers due to confusion or drift. The current truncation approach and pilot study seem to overlook this dimension. (W2) | 2. We have further extended pilot studies (Figure 1 c.) to examine the effect of truncating incorrect answers on accuracy and output token length, providing more detailed support for the motivation behind JET’s design. | |
|  | 3. Effect of PES without the length reward and Comparison with established length reward baselines. (W3) | 3. We have conducted more detailed ablation studies on the length reward in Section 4.7, including: removing the length reward to analyze its contribution (Appendix C.4), applying the length reward to all answers in a group to further demonstrate the effectiveness of our length reward mechanism (Appendix C.5), and comparing with length reward designs from other works to highlight the advantage of ours within JET (Section 4.7). | |

We are sincerely grateful to all reviewers for their constructive feedback, which has greatly helped improve the quality and completeness of our work and made the technical contributions more solid. We would like to give **special thanks to Reviewers Zu3p and tagM** for their positive engagement and recognition of our efforts during the rebuttal period, as reflected in their score increases. We also sincerely appreciate the **original AC for his/her support**, which encouraged the reviewers to actively engage in discussion. We truly value and appreciate the reviewers’ and AC’s contributions to enhancing this work and supporting the research community.

---

### Meta-Review · Area_Chair_S1MD · 2026-01-22

**Summary:**

This paper tackles the “overthinking” inefficiency of Large Reasoning Models by proposing Just-Enough Thinking (JET), an RL-based training approach that encourages models to terminate reasoning earlier without sacrificing correctness. The core idea is to generate short, distributionally consistent reasoning trajectories by truncating model-produced rollouts (two-stage rollout with Progressive Early-Stopping, PES) and to apply a quality-controlled length reward among correct solutions to prefer shorter reasoning. Reviewers generally agree the motivation is timely and important, and that the pilot experiments support the premise that models often accumulate sufficient evidence early. The work is technically plausible and mostly well presented.

The main points of contention in the initial reviews were: (1) whether the truncation + stop cue introduces harmful distribution mismatch, (2) whether JET is clearly superior to strong baselines such as Laser(-DE) (especially on the 7B backbone), and (3) limited evaluation beyond DeepSeek-Distill-Qwen. In the rebuttal, the authors added theoretical justification and empirical training dynamics (KL) to address distributional inconsistency, expanded ablations on reward weights and the contribution of the length reward, added discussion/analysis comparing to Laser, and introduced additional backbone evidence (DeepSeek-R1-Distill-Llama-8B). Overall, the rebuttal meaningfully improves clarity and addresses most of the reviewers’ major concerns.

**Reviewer Concerns:**

**Addressed by the Rebuttal**

1. ***Distributional inconsistency / “naturalness” of truncated rollouts (Zu3p, 6BQF):*** The authors provide a clearer rationale for why truncation stays close to the policy distribution (truncate only model-generated trajectories; minimal stop tokens) and add empirical evidence via KL-loss dynamics showing JET does not diverge more than DAPO.

2. ***Limited backbone evaluation (Zu3p, 6BQF, tagM):*** The authors added experiments on DeepSeek-R1-Distill-Llama-8B, demonstrating similar accuracy. This addition reduces the “Qwen-only” risk.

3. ***Ablations and design justification (A4Dw, tagM):*** The rebuttal adds analyses on reward-weight sensitivity and an ablation indicating the length reward is important beyond PES alone. Ambiguities in definitions were clarified.

4. ***Overthinking leading to incorrect answers (tagM):*** The authors extended pilot analysis to cases where truncation can correct initially wrong long trajectories (error-correction rate), partially addressing the broader overthinking definition.

**Partially Addressed by the Rebuttal**

1. ***Comparison to Laser and “sub-optimal” 7B results (Zu3p, 6BQF):*** The authors argue JET offers more stable behavior across difficulties, stronger out-of-domain generalization, and lower training cost / simpler implementation. This is helpful, but the paper still has a mild weakness: on some in-domain math settings (notably 7B), Laser variants remain competitive or better on certain points of the Pareto frontier. The revised discussion reduces confusion, but the empirical gap remains.

2. ***Limitations / failure modes (6BQF):*** The authors add a brief statement that tasks with inherently dense necessary reasoning may offer limited compressibility. This is reasonable but still somewhat light; a clearer limitation section would strengthen the final paper.

**Reviewer Scores:**

1. Reviewer Zu3p (initial: 4, “marginally below threshold”): Explicitly stated in discussion that concerns are addressed and they intended to increase the score but were blocked by system restrictions. I would estimate +1 (to around 5: marginally above threshold) based on their final comment.

2. Reviewer tagM (final updated score: 6): They updated to 6 after rebuttal; this indicates clear improvement in their view, especially on evaluation breadth and missing ablations.

3. Reviewer 6BQF (initial: 4): Many concerns overlap with Zu3p (Laser comparison, distributional mismatch, backbone breadth, training time). The rebuttal addresses most of these with added KL analysis, new backbone, and explicit training-time comparisons; I would expect a moderate increase, likely +1 (to 5), though not guaranteed.

4. Reviewer A4Dw (initial: 4): The authors responded to the reward-weight question, clarified non-upper-bound reasoning vs DAPO, clarified annotation, and provided some analysis. Remaining skepticism (e.g., “penalizing longer correct answers”) is partly philosophical but addressed. Likely 0 to +1, I would conservatively estimate +0.5.

---

### Decision · Program_Chairs · 2026-01-26

Accept (Poster)